# BILBO: BILEVEL BAYESIAN OPTIMIZATION

## ABSTRACT

Bilevel optimization, characterized by a two-level hierarchical optimization structure, is prevalent in real-world problems but poses significant challenges, especially in noisy, constrained, and derivative-free settings. To tackle these challenges, we present a novel algorithm for BILevel Bayesian Optimization (BILBO) that optimizes both upper- and lower-level problems jointly in a sample-efficient manner by using confidence bounds to construct trusted sets of feasible and lower-level optimal solutions. We show that sampling from our trusted sets guarantees points with instantaneous regret bounds. Moreover, BILBO selects only one function query per iteration, facilitating its use in decoupled settings where upper- and lower-level function evaluations may come from different simulators or experiments. We also show that this function query selection strategy leads to an instantaneous regret bound for the query point. The performance of BILBO is theoretically guaranteed with a sublinear regret bound and is empirically evaluated on several synthetic and real-world problems.

## 1 INTRODUCTION

Many real-world problems have hierarchical decision making processes involving two levels of optimization. Decisions made at the upper level affect the optimization problem at the lower level and vice versa. Bilevel optimization can model hierarchical structures well and enable analysis of such problems. Applications of bilevel optimization range from machine learning (e.g., hyperparameter optimization, meta-learning) to economic problems (e.g., pricing strategies, toll setting) (Beck & Schmidt, 2021). In energy management, energy providers determine optimal pricing strategies for electricity (upper level) while consumers optimize their electricity demands based on the pricing (lower level). Similarly, in investment, brokers or regulators set fees on different asset classes to maximize their revenues (upper level), while investors optimize their portfolios considering expected returns and risk (lower level). Bilevel optimization has been applied in both cases (Shu et al., 2018; Leal et al., 2020), typically using a nested framework with linear solvers at the lower level. This approach may limit practical effectiveness but it is due to the inherent complexity of bilevel optimization. Even with only linear constraints and objective functions, the set of feasible solutions can be non-convex and non-continuous (Kleinert et al., 2021). Lower-level solutions that are $\epsilon$-feasible w.r.t. non-linear constraints may also lead to a bilevel solution that is arbitrarily far from the actual bilevel solution (Beck et al., 2023).

Classical approaches (Bard & Falk, 1982; Bard & Moore, 1990) have relied on simplifying assumptions, such as linearity or convexity, while others, assuming the presence of gradients, use gradient descent to solve the lower-level problem and approximate hypergradients for the upper level. On the other hand, meta-modeling-based methods employ surrogate models for efficient optimization, like BLEAQ (Sinha et al., 2013) which uses quadratic approximations to map upper-level points to optimal lower-level solutions. Meta-modeling methods have advantages over gradient-based methods in the presence of noisy observations, derivative-free functions, constraints and discrete variables. They can also enable global optimization, even when dealing with non-convex functions. However, existing meta-modeling methods are limited by the fit of low capacity models and lack of theoretical analysis on the performance bounds.

Bayesian optimization (BO), a popular meta-modeling method, has been applied extensively, including to constrained optimization (Gelbart et al., 2014). In particular, Xu et al. (2023) and Nguyen et al. (2023) both introduced confidence-bounds based optimistic estimations of the feasible set, with the former providing an infeasibility declaration scheme and the latter including a function

query strategy for decoupled settings. Compared to constrained optimization, bilevel optimization has additional significant challenges from being constrained by unknown optimal lower-level solutions, which requires optimization of a separate lower-level problem. These challenges also differ from those in robust optimization (Bogunovic et al., 2018) and composite objectives optimization (Li & Scarlett, 2022), as both are single level optimization, despite the additional random variable and composite objective function respectively. Our work tackles challenges in bilevel optimization, particularly where we consider the optimality of estimated lower-level solutions.

In bilevel optimization, BO has only been employed in a nested framework (Kieffer et al., 2017; Dogan & Prestwich, 2023). This approach involves optimizing the upper level via BO while separately optimizing the lower level at each upper-level query point. The nested approach suffers from sample inefficiency due to the lack of information flow between the upper- and lower-levels of the optimization problem. To address this issue, Dogan & Prestwich (2023) introduced an acquisition function that is conditional on the optimal points of the lower level during the upper-level optimization. It still requires the lower-level problem to be optimized separately to convergence at each upper-level point. On theoretical analysis, Fu et al. (2024) provided a theoretical guarantee for a bilevel framework with stochastic gradient descent at the lower-level and BO at the upper-level. However, the lower-level is not a blackbox optimization due to this gradient assumption.

**Contributions.** We propose BILevel Bayesian Optimization (BILBO), a Bayesian optimization algorithm for general bilevel problems with blackbox functions, where both levels are optimized simultaneously. BILBO introduces trusted sets to iteratively reduce the search space by removing infeasible solutions and sub-optimal lower-level solutions. Points in the trusted sets have upper-bounded instantaneous regrets, ensuring that they are good candidates for sampling. Functions are modelled using Gaussian processes (GPs) over both upper- and lower-level variables, which improves sample efficiency by enhancing the flow of information between lower-level optimization and various upper-level points. The trusted sets are constructed from confidence bounds of GPs, enabling the derivation of regret bounds. Intuitively, the trusted sets provide estimates of feasible solutions for upper-level optimization without requiring the lower-level problem to be optimized to convergence. We also address the decoupled setting, where only one function is chosen to be queried per iteration instead of querying all the functions at both levels simultaneously. This approach allows different functions of varying complexities or costs to have different number of evaluations for a good approximation. We show that our function query selection strategy provides an instantaneous regret bound for the query point, which leads to a sublinear regret bound. The code and data used in the paper will be released upon publication. For clarity, notations used in this paper are summarized in a notation table provided in Appendix A.

## 2 PRELIMINARIES

**Bilevel optimization.** Let $F$ and $f$, respectively, be the upper- and lower-level black-box objective function, such that $F, f : \mathcal{X} \times \mathcal{Z} \to \mathbb{R}$. Let $\mathcal{C}_{\text{up}}, \mathcal{C}_{\text{lo}}$ respectively, be sets of upper- and lower-level black-box constraints where $c : \mathcal{X} \times \mathcal{Z} \to \mathbb{R}, \ \forall c \in \mathcal{C}_{\text{up}} \cup \mathcal{C}_{\text{lo}}$. The upper-level variable is denoted as $\mathbf{x} \in \mathcal{X}$ and lower-level variable as $\mathbf{z} \in \mathcal{Z}$, where $\mathcal{X} \subset \mathbb{R}^{d_{\mathcal{X}}}$ and $\mathcal{Z} \subset \mathbb{R}^{d_{\mathcal{Z}}}$ are assumed to be finite. We consider a general bilevel optimization problem with constraints as

$$\max_{\mathbf{x} \in \mathcal{X}, \mathbf{z} \in \mathcal{P}(\mathbf{x})} F(\mathbf{x}, \mathbf{z}) \tag{2.1}$$

$$\text{s.t.} \quad C(\mathbf{x}, \mathbf{z}) \geq 0, \ \ \forall C \in \mathcal{C}_{\text{up}}, \tag{2.2}$$

$$\mathcal{P}(\mathbf{x}) \triangleq \{\arg\max_{\mathbf{z} \in \mathcal{Z}} f(\mathbf{x}, \mathbf{z}) \mid c(\mathbf{x}, \mathbf{z}) \geq 0, \ \ \forall c \in \mathcal{C}_{\text{lo}}\} \tag{2.3}$$

where $\mathcal{P}(\mathbf{x})$ is the set of optimal lower-level solutions at upper-level variable $\mathbf{x}$.

Let $(\mathbf{x}^*, \mathbf{z}^*)$ denote the optimal bilevel solution, and $(\mathbf{x}, \mathbf{z}^*(\mathbf{x}))$ denote the optimal lower-level solution w.r.t. $\mathbf{x}$, where $\mathbf{z}^* = \mathbf{z}^*(\mathbf{x}^*)$. The set of functions $\mathcal{F}$ is defined as $\mathcal{F} \triangleq \{F, f\} \cup \mathcal{C}_{\text{up}} \cup \mathcal{C}_{\text{lo}}$. At each step $t \geq 1$, we select a query point $(\mathbf{x}_t, \mathbf{z}_t)$ and obtain noisy observations $y_h(\mathbf{x}_t, \mathbf{z}_t) \triangleq h(\mathbf{x}_t, \mathbf{z}_t) + \epsilon$ where $\epsilon \sim \mathcal{N}(0, \sigma_n^2), \forall h \in \mathcal{F}$. In a decoupled setting, a function query $h_t \in \mathcal{F}$ is selected, and only $y_{h_t}(\mathbf{x}_t, \mathbf{z}_t)$ is observed. Observations are accumulated into $\mathcal{D}_{h_t, t} \triangleq \mathcal{D}_{h_t, t-1} \cup \{y_{h_t}(\mathbf{x}_t, \mathbf{z}_t)\}$ and $\mathcal{D}_{h,0}$ is the set of initial observations for function $h$. To approximate the optimal solution $(\mathbf{x}^*, \mathbf{z}^*)$, $(\mathbf{x}_t, \mathbf{z}_t)$ is commonly used as an estimator.

**Gaussian process.** Each function $h \in \mathcal{F}$ is modelled with a Gaussian process (GP). Let $\mathbf{xz}$ be a concatenation of $\mathbf{x}$ and $\mathbf{z}$. A $\mathcal{GP}_h(m_h(\mathbf{xz}), k_h(\mathbf{xz}, \mathbf{xz}'))$ is specified by a mean function $m_h(\mathbf{xz}) \triangleq \mathbb{E}[h(\mathbf{xz})]$ and covariance function $k_h(\mathbf{xz}, \mathbf{xz}') \triangleq \mathbb{E}[(h(\mathbf{xz}) - m_h(\mathbf{xz}))(h(\mathbf{xz}') - m_h(\mathbf{xz}'))]$ (Williams & Rasmussen, 2006). At iteration $t$, given query inputs $\mathbf{xz}_{:t-1}$ and noisy observations $\mathbf{y}_{h,t-1}$, the predictive distribution for $h$ is Gaussian: $h(\mathbf{xz}) \mid \mathbf{xz}_{:t-1}, \mathbf{y}_{h,t-1} \sim \mathcal{N}(\mu_{h,t-1}(\mathbf{xz}), \sigma_{h,t-1}^2(\mathbf{xz}))$. The closed-form posteriors can be found in Appendix B.1.

We introduce the maximum information gain from Srinivas et al. (2010) for a function $h$ as $\gamma_{h,t} \triangleq \max_{\{(\mathbf{x}_t, \mathbf{z}_t)\}_{t \in T(h)}} \frac{1}{2} \log |\mathbf{I} + \sigma^{-2} \mathbf{K}_{h,t}|$, where $T(h)$ contains the timesteps where function $h$ was selected for query. $\gamma_{h,t}$ for common kernels were found to be sublinear and shown in Appendix B.2.

**Bayesian optimization.** Given a prior distribution $P(h)$ and likelihood function $P(\mathcal{D}_{h_t,t}|h)$, the posterior distribution $P(h|\mathcal{D}_{h_t,t})$ can be calculated via Bayes' theorem. The prior distribution is often represented by a GP and the likelihood function is defined by the choice of GP kernel and hyperparameters. The posterior distribution is also a surrogate function for modeling $h$. The point which maximizes an acquisition function $a_h(\mathbf{xz})$ is selected as the next point to evaluate function $h$ at. A popular acquisition strategy is based on confidence bounds (Srinivas et al., 2010).

**Regrets.** Regret is defined as the loss in reward from not selecting the optimal point. Instantaneous regret $r_t$ measures this loss at time $t$, while cumulative regret $R_T \triangleq \sum_{t=1}^{T} r_t$ is the sum of instantaneous regrets over $T$ rounds. An algorithm is no-regret if $\lim_{T \to \infty} R_T/T = 0$ where cumulative regret is sublinear and convergence to the optimal point is guaranteed with a large enough $T$.

For bilevel optimization, we propose to define the instantaneous regret as

$$r_t \triangleq \max_{h \in \mathcal{F}} r_h(\mathbf{x}_t, \mathbf{z}_t), \tag{2.4}$$

where $\mathcal{F} \triangleq \{F, f\} \cup \mathcal{C}_{\text{up}} \cup \mathcal{C}_{\text{lo}}$. The upper- and lower-level instantaneous objective regrets are defined, respectively, as

$$r_F(\mathbf{x}_t, \mathbf{z}_t) \triangleq F(\mathbf{x}^*, \mathbf{z}^*) - F(\mathbf{x}_t, \mathbf{z}_t), \tag{2.5}$$

$$r_f(\mathbf{x}_t, \mathbf{z}_t) \triangleq f(\mathbf{x}_t, \mathbf{z}^*(\mathbf{x}_t)) - f(\mathbf{x}_t, \mathbf{z}_t), \tag{2.6}$$

and the instantaneous constraint regrets as

$$\forall c \in \mathcal{C}_{\text{up}} \cup \mathcal{C}_{\text{lo}} \quad r_c(\mathbf{x}_t, \mathbf{z}_t) \triangleq \max(0, -c(\mathbf{x}_t, \mathbf{z}_t)). \tag{2.7}$$

Note that $r_c$ is usually known as constraint violations and we have incorporated them into the definition of bilevel optimization regret to have a more representative estimate of the optimality of a point. An algorithm that is no-regret according to our definition of bilevel optimization regret, will have all objective function regrets and constraint violations converge to $0$. If the constraints and objective functions have very different ranges, normalization of the output values can ensure a fairer representation of the overall regret.

## 3 BILEVEL BAYESIAN OPTIMIZATION

Our method, called *BILevel Bayesian Optimization* (BILBO), maintains and samples from trusted sets constructed from confidence bounds, where points in the trusted sets have upper-bounded instantaneous regrets on constraints and the lower-level objective. This ensures that points in the trusted sets are sufficiently good for use in the optimization of the upper-level problem, enhancing sample efficiency. We introduce a function query strategy based on estimated regrets, which provides an instantaneous regret bound on the query point. This leads to a sublinear cumulative regret bound for BILBO. The pseudocode is in Algorithm 1.

The trusted sets are defined in Definitions 3.3 and 3.5, and Lemmas 3.4 and 3.6 provide instantaneous regret bounds on points in the trusted sets. Definition 3.7 defines the function query selection and Lemma 3.8 presents a instantaneous regret bound on the query point. Finally, the cumulative regret bound of BILBO is stated in Theorem 3.9 and the simple regret bound in Lemma 3.10.

First, we define the confidence bounds on which we use to build the trusted sets. Functions are bounded by the confidence bounds with high probability by Corollary 3.2.

**Definition 3.1** (Confidence bounds). For a function $h \in \mathcal{F}$ modelled by a Gaussian process (GP), $\forall \mathbf{x} \in \mathcal{X}, \mathbf{z} \in \mathcal{Z}$, and $t \geq 1$, let the upper and lower confidence bounds of $h(\mathbf{x}, \mathbf{z})$ be denoted, respectively, as

$$u_{h,t}(\mathbf{x}, \mathbf{z}) \triangleq \mu_{h,t-1}(\mathbf{x}, \mathbf{z}) + \beta_t^{1/2} \sigma_{h,t-1}(\mathbf{x}, \mathbf{z}), \tag{3.1}$$

$$l_{h,t}(\mathbf{x}, \mathbf{z}) \triangleq \mu_{h,t-1}(\mathbf{x}, \mathbf{z}) - \beta_t^{1/2} \sigma_{h,t-1}(\mathbf{x}, \mathbf{z}), \tag{3.2}$$

where $\mu_{h,t-1}(\mathbf{x}, \mathbf{z})$ and $\sigma_{h,t-1}(\mathbf{x}, \mathbf{z})$ are the GP's posterior mean and standard deviation at $(\mathbf{x}, \mathbf{z})$, and $\beta_t \triangleq 2 \log(|\mathcal{F}||\mathcal{X}||\mathcal{Z}|t^2 \pi^2 / (6\delta))$.

**Corollary 3.2.** *For some small $\delta > 0$, with probability at least $1 - \delta$,*

$$h(\mathbf{x}, \mathbf{z}) \in [l_{h,t}(\mathbf{x}, \mathbf{z}), u_{h,t}(\mathbf{x}, \mathbf{z})].$$

*This is derived from Lemma 5.1 of Srinivas et al. (2010) by applying union bound over $h \in \mathcal{F}$.*

---

**Algorithm 1** BILBO

---

**Require:** $\mathcal{X}, \mathcal{Z}, \{\mathcal{D}_{h,0}\}_{h \in \mathcal{F}}$
  1: Update GP posterior beliefs: $\{(\mu_{h,0}, \sigma_{h,0})\}_{h \in \mathcal{F}}$
  2: Update trusted sets $\mathcal{S}_t^+, \mathcal{P}_t^+$                           ▷ Definitions 3.3 and 3.5
  3: **for** $t \leftarrow 1$ to $T$ **do**
  4:     **if** $\mathcal{S}_t^+ = \emptyset$ **then**
  5:         Declare infeasibility
  6:     **end if**
  7:     $\mathbf{x}_t, \mathbf{z}_t \leftarrow \arg\max_{(\mathbf{x}, \mathbf{z}) \in \mathcal{S}_t^+ \cap \mathcal{P}_t^+} u_{F,t}(\mathbf{x}, \mathbf{z})$        ▷ Equation 3.6
  8:     $h_t \leftarrow \arg\max_{h \in \mathcal{F}} \bar{r}_{h,t}(\mathbf{x}_t, \mathbf{z}_t)$              ▷ Definition 3.7
  9:     **if** $h_t = f$ **then**
10:         $\bar{\mathbf{z}}_t \leftarrow \arg\max_{\mathbf{z} \in \mathcal{S}_{\mathrm{lo},t}^+(\mathbf{x}_t)} u_{f,t}(\mathbf{x}_t, \mathbf{z})$       ▷ Equation 3.5
11:         **if** $\sigma_{f,t-1}(\mathbf{x}_t, \bar{\mathbf{z}}_t) > \sigma_{f,t-1}(\mathbf{x}_t, \mathbf{z}_t)$ **then**
12:             $\mathbf{z}_t \leftarrow \bar{\mathbf{z}}_t$                          ▷ Equation 3.8
13:         **end if**
14:     **end if**
15:     $\mathcal{D}_{h_t,t} \leftarrow \mathcal{D}_{h_t,t-1} \cup \{y_{h_t}(\mathbf{x}_t, \mathbf{z}_t)\}$
16:     Update GP posterior belief: $\mu_{h_t,t}, \sigma_{h_t,t}$
17:     Update trusted sets $\mathcal{S}_t^+, \mathcal{P}_t^+$              ▷ Definitions 3.3 and 3.5
18: **end for**

---

### 3.1 TRUSTED SET OF FEASIBLE SOLUTIONS

The optimal solution must not violate any of the constraints present. To approximate the unknown feasible regions, we introduce a trusted set of feasible solutions using confidence bounds.

**Definition 3.3** (Trusted set of feasible solutions). Let the trusted set of feasible solutions be defined as

$$\mathcal{S}_t^+ \triangleq \{(\mathbf{x}, \mathbf{z}) \in \mathcal{X} \times \mathcal{Z} \mid u_{c,t}(\mathbf{x}, \mathbf{z}) \geq 0 \ \forall c \in \mathcal{C}_{\mathrm{up}} \cup \mathcal{C}_{\mathrm{lo}}\}, \tag{3.3}$$

where $\mathcal{C}_{\mathrm{up}} \cup \mathcal{C}_{\mathrm{lo}}$ is the set of all constraints, and the upper confidence bound $u_{c,t}$ is defined in Definition 3.1. For convenience, let $\mathcal{S}_t^+(\mathbf{x}) \triangleq \{\mathbf{z} \mid (\mathbf{x}, \mathbf{z}) \in \mathcal{S}_t^+\}$.

**Lemma 3.4.** $\forall (\mathbf{x}, \mathbf{z}) \in \mathcal{S}_t^+, c \in \mathcal{C}_{up} \cup \mathcal{C}_{lo}$, *the constraint regrets are upper bounded,*

$$r_c(\mathbf{x}, \mathbf{z}) \leq 2\beta_t^{1/2} \sigma_{c,t-1}(\mathbf{x}, \mathbf{z}).$$

The proof is provided in Appendix C.1.

Selecting a query point from $\mathcal{S}_t^+$ ensures the instantaneous constraint regrets of the chosen point is upper bounded, where highly infeasible points are outside of the trusted set. An empty trusted feasible set would imply an infeasible bilevel problem, and our algorithm would make an infeasibility declaration.

## 3.2 TRUSTED SET OF OPTIMAL LOWER-LEVEL SOLUTIONS

The upper-level problem is also constrained by the set of optimal lower-level solutions, and we define a trusted set of optimal lower-level solutions to approximate the unknown optimal lower-level solutions.

**Definition 3.5** (Trusted set of optimal lower-level solutions). Let the trusted set of optimal lower-level solutions be defined as

$$\mathcal{P}_t^+ \triangleq \{(\mathbf{x}, \mathbf{z}) \in \mathcal{S}_{\mathrm{lo},t}^+ \mid u_{f,t}(\mathbf{x}, \mathbf{z}) \geq l_{f,t}(\mathbf{x}, \bar{\mathbf{z}}_t(\mathbf{x}))\}, \tag{3.4}$$

where $\mathcal{S}_{\mathrm{lo},t}^+ \triangleq \{(\mathbf{x}, \mathbf{z}) \in \mathcal{X} \times \mathcal{Z} \mid u_{c,t}(\mathbf{x}, \mathbf{z}) \geq 0 \ \ \forall c \in \mathcal{C}_{\mathrm{lo}}\}$ is the trusted set of feasible solutions w.r.t. lower-level constraints, and

$$\bar{\mathbf{z}}_t(\mathbf{x}) \triangleq \arg \max_{\mathbf{z} \in \mathcal{S}_{\mathrm{lo},t}^+(\mathbf{x})} u_{f,t}(\mathbf{x}, \mathbf{z}) \tag{3.5}$$

is the estimated optimal lower-level solution at $\mathbf{x}$.

The trusted set $\mathcal{P}_t^+$ allows multiple lower-level solutions to correspond to an upper-level variable, enabling our algorithm to effectively manage multiple lower-level optima and noisy observations. Moreover, we can handle infeasible lower-level problems, as the trusted set $\mathcal{P}_t^+$ naturally filters out highly probable infeasible points via the set $\mathcal{S}_{\mathrm{lo},t}^+$, including all points with an infeasible lower-level problem, ensuring that only probable feasible solutions are considered during optimization.

**Lemma 3.6.** $\forall (\mathbf{x}, \mathbf{z}) \in \mathcal{P}_t^+$, *the lower-level objective regret is upper bounded,*

$$r_{f,t}(\mathbf{x}, \mathbf{z}) \leq \mathbb{1}_{\mathbf{z} \neq \bar{\mathbf{z}}_t(\mathbf{x})} 2\beta_t^{1/2} \sigma_{f,t-1}(\mathbf{x}, \bar{\mathbf{z}}_t(\mathbf{x})) + 2\beta_t^{1/2} \sigma_{f,t-1}(\mathbf{x}, \mathbf{z}).$$

The proof is provided in Appendix C.2.

Sampling from $\mathcal{P}_t^+$ guarantees an upper-bounded lower-level objective regret, and points outside of the trusted set $\mathcal{P}_t^+$ are highly unlikely to be lower-level optimal.

$\epsilon$**-optimal lower-level solutions**. In some scenarios, it may be desirable to consider $\epsilon$-optimal lower-level solutions feasible, as it is common for real-world agents to operate sub-optimally. This approach allows us to account for practical limitations where perfect lower-level optimization may not be achievable, for example, due to noise or the expense of querying the lower-level function. In this case, we can relax the condition in Definition 3.5 to allow $\epsilon$-optimal lower-level solutions to remain in the trusted set by defining $\mathcal{P}_t^\epsilon \triangleq \{(\mathbf{x}, \mathbf{z}) \mid u_{f,t}(\mathbf{x}, \mathbf{z}) + \epsilon \geq l_{f,t}(\mathbf{x}, \bar{\mathbf{z}}_t(\mathbf{x}))\}$, and extending the regret bound in Lemma 3.6 to $r_{f,t}(\mathbf{x}, \mathbf{z}) \leq \epsilon + \mathbb{1}_{\mathbf{z} \neq \bar{\mathbf{z}}_t(\mathbf{x})} 2\beta_t^{1/2} \sigma_{f,t-1}(\mathbf{x}, \bar{\mathbf{z}}_t(\mathbf{x})) + 2\beta_t^{1/2} \sigma_{f,t-1}(\mathbf{x}, \mathbf{z})$.

## 3.3 QUERY POINT SELECTION

We reduce the search space to $\mathcal{S}_t^+ \cap \mathcal{P}_t^+$. Points in this search space have upper-bounded instantaneous regrets on constraints and lower-level objective with high probability, according to Lemmas 3.4 and 3.6. The query point at timestep $t$ is sampled from the reduced search space and chosen at the maximum upper confidence bound of upper-level objective $u_{F,t}$,

$$\mathbf{x}_t, \mathbf{z}_t \triangleq \arg \max_{(\mathbf{x}, \mathbf{z}) \in \mathcal{S}_t^+ \cap \mathcal{P}_t^+} u_{F,t}(\mathbf{x}, \mathbf{z}) . \tag{3.6}$$

## 3.4 FUNCTION QUERY

In the decoupled case, a function query $h_t$ is selected at each timestep $t$ for evaluation. We follow the function query selection in Definition 3.7, and Lemma 3.8 provides an instantaneous regret bound on the query $(\mathbf{x}_t, \mathbf{z}_t)$.

**Definition 3.7** (Function query). Let the function query $h_t$ selected at each timestep $t$ be

$$h_t \triangleq \arg \max_{h \in \mathcal{F}} \bar{r}_{h,t}(\mathbf{x}_t, \mathbf{z}_t), \tag{3.7}$$

where $\mathcal{F} \triangleq \{F, f\} \cup \mathcal{C}_{\mathrm{up}} \cup \mathcal{C}_{\mathrm{lo}}$, and the estimated regrets are defined as

$$\forall h' \in \mathcal{F}/\{f\}, \ \bar{r}_{h',t}(\mathbf{x}_t, \mathbf{z}_t) \triangleq 2\beta_t^{1/2} \sigma_{h',t-1}(\mathbf{x}_t, \mathbf{z}_t),$$

$$\bar{r}_{f,t}(\mathbf{x}_t, \mathbf{z}_t) \triangleq \mathbb{1}_{\mathbf{z} \neq \bar{\mathbf{z}}_t(\mathbf{x}_t)} 2\beta_t^{1/2} \sigma_{f,t-1}(\mathbf{x}_t, \bar{\mathbf{z}}_t(\mathbf{x}_t)) + 2\beta_t^{1/2} \sigma_{f,t-1}(\mathbf{x}_t, \mathbf{z}_t),$$

where $\bar{\mathbf{z}}_t$ is defined in Equation 3.5.

**Reassignment of $\mathbf{z}_t$ for lower-level objective function query.** The lower-level variable to query, $\mathbf{z}_t$, has to be reassigned as follows,

$$\mathbf{z}_t \leftarrow \bar{\mathbf{z}}_t(\mathbf{x}_t) \ \text{ if } h_t = f \text{ and } \sigma_{f,t-1}(\mathbf{x}_t, \bar{\mathbf{z}}_t(\mathbf{x}_t)) \geq \sigma_{f,t-1}(\mathbf{x}_t, \mathbf{z}_t). \tag{3.8}$$

Intuitively, we want to reduce the estimated regret $\bar{r}_{f,t}(\mathbf{x}_t, \mathbf{z}_t)$, which comprises both $\sigma_{f,t-1}(\mathbf{x}_t, \bar{\mathbf{z}}_t(\mathbf{x}_t))$ and $\sigma_{f,t-1}(\mathbf{x}_t, \mathbf{z}_t)$ terms. Reassigning $\mathbf{z}_t$ to the term that contributes more to $\bar{r}_t(\mathbf{x}_t, \mathbf{z}_t)$ reduces the estimated regret more effectively. If $f$ is only queried at $(\mathbf{x}_t, \mathbf{z}_t)$, $\sigma_{f,t-1}(\mathbf{x}_t, \bar{\mathbf{z}}_t(\mathbf{x}_t))$ would remain large even after repeated queries to $f$. This reassignment is integral to manage the uncertainty of estimated lower-level solutions as we sample query points from the trusted set $\mathcal{P}_t^+$ and do not globally optimize the lower-level problem at any upper-level point.

**Lemma 3.8.** *Following the function query selection in Definition 3.7 and reassignment of query point in Equation 3.8, the instantaneous regret for the query point $(\mathbf{x}_t, \mathbf{z}_t)$ at time $t \geq 1$ is upper bounded by,*

$$r_t \leq 4\beta_t^{1/2} \max_{h \in \mathcal{F}} \sigma_{h,t-1}(\mathbf{x}_t, \mathbf{z}_t).$$

The proof is in Appendix C.3. By Lemma C.4, we also see that $\max_{h \in \mathcal{F}} \bar{r}_{h,t}(\mathbf{x}_t, \mathbf{z}_t) \geq r_t$. Thus, $\max_{h \in \mathcal{F}} \bar{r}_{h,t}(\mathbf{x}_t, \mathbf{z}_t)$ can be interpreted as the upper regret bound at query point $(\mathbf{x}_t, \mathbf{z}_t)$ where a large $\bar{r}_{h,t}(\mathbf{x}_t, \mathbf{z}_t)$ suggests that function $h$ affects $r_t$ significantly. Since $\bar{r}_{h,t}$ comprises of $\sigma_{h,t-1}$, selecting the $\arg\max_{h \in \mathcal{F}} \bar{r}_{h,t}$ in Definition 3.7 can also be seen as selecting the most uncertain function to query at $(\mathbf{x}_t, \mathbf{z}_t)$.

## 3.5 REGRET BOUND

The cumulative regret of Algorithm 1 is shown in Theorem 3.9 and proven in Appendix C.4 using Lemma 3.8.

**Theorem 3.9.** *Let $\delta \in (0, 1)$ and $\beta_t \triangleq 2\log(|\mathcal{F}||\mathcal{X}||\mathcal{Z}|t^2\pi^2/6\delta)$. With probability of at least $1 - \delta$, Algorithm 1 has a cumulative regret bound of*

$$R_T \leq \sqrt{4T|\mathcal{F}|\beta_T \max_{h \in \mathcal{F}} C_h \gamma_{h,T}},$$

*where $C_h \triangleq 8/\log(1 + \sigma_h^{-2})$, and $\gamma_{h,T}$ is the maximum information gain from noisy observations of $h$ at $(\mathbf{x}_t, \mathbf{z}_t) \ \forall t \in [T]$.*

This indicates that the regret bound is related to the maximum information gain across all functions in $\mathcal{F}$. Specifically, the term $\max_{h \in \mathcal{F}}$ in our definition of bilevel regret in Equation 2.4 contributed to this relationship. The presence of this maximum term suggests that the overall regret is influenced by the most challenging function within the set $\mathcal{F}$.

Our regret bound includes a larger constant compared to the regret bound for constrained Bayesian optimization in Nguyen et al. (2023), which reflects the increased difficulties in optimizing bilevel problems. The regret arising from the lower-level objective has a larger upper bound than regrets from other constraints, indicating that sub-optimal lower-level solutions have a more significant impact on upper-level optimization, making the optimization process more complex than standard constrained optimization.

The cumulative regret bound of BILBO is sublinear as $\gamma_{h,T}$ is sublinear for common kernels including Squared Exponential and Matérn kernels (Srinivas et al., 2010). The sublinear cumulative regret guarantees convergence to the optimal solution as $R_T/T \to 0$ as $T \to \infty$.

Selecting an estimator as

$$\hat{\mathbf{x}}_T, \hat{\mathbf{z}}_T \triangleq \arg \min_{(\mathbf{x}_t, \mathbf{z}_t) \in \{(\mathbf{x}_{t'}, \mathbf{z}_{t'})\}_{t' \in [T]}} \max_{h \in \mathcal{F}} \bar{r}_{h,t}(\mathbf{x}_t, \mathbf{z}_t), \tag{3.9}$$

we provide a simple regret bound in Lemma 3.10.

**Lemma 3.10.** *With probability at least $1 - \delta$, $T \geq 1$, the estimator $(\hat{\mathbf{x}}_T, \hat{\mathbf{z}}_T)$, defined in Equation 3.9, has a simple regret bound of*

$$r_T \leq \sqrt{4|\mathcal{F}|\beta_T \max_{h \in \mathcal{F}} C_h \gamma_{h,T}/T},$$

*where $\beta_T \triangleq 2\log(|\mathcal{F}||\mathcal{X}||\mathcal{Z}|T^2\pi^2/6\delta)$.*

This follows as the simple regret of $(\hat{\mathbf{x}}_T, \hat{\mathbf{z}}_T)$ is upper bounded by the average regret bound in Lemma 3.8 across timesteps. The detailed proof is in Appendix C.5.

## 4 EXPERIMENTS

We evaluate the performance of BILBO on 4 synthetic and 2 real-world problems. We compare BILBO with 2 baselines we introduced: "TrustedRand" and "Nested". TrustedRand involves randomly sampling query points from trusted sets, which can provide valuable insights into how much trusted sets contribute to the overall performance of BILBO. On the other hand, Nested optimizes the upper- and lower-level problems separately, which serves as a baseline for nested BO approaches such as those in Kieffer et al. (2017); Dogan & Prestwich (2023). More details on TrustedRand and Nested are in Appendix D.1. Note that Nested cannot handle constraints and it is not compared in experiments with constraints.

Algorithms are implemented using GpyTorch (Gardner et al., 2018). All experiments, except Nested, are initialized with 3 observations on each function. Nested approaches require more initial observations of the lower-level functions because the upper-level objective is only evaluated at the estimated optimal lower-level solution. We allow for this to enable comparisons, which also highlights the sample inefficiency of nested methods. All observations are noisy with $\sigma_n$ set to 0.01, and the outputs are normalized to have a mean of 0 and a standard deviation of 1. We discretize the search space using a uniformly-spaced grid in our implementation of BO to facilitate representation of trusted sets. BILBO is implemented in a decoupled way where we query only one function per iteration, while TrustedRand queries all function at each iteration. For comparison, the estimator is chosen as $\arg\max_{(\mathbf{x},\mathbf{z})\in\mathcal{S}_t^+\cap\mathcal{P}_t^+}\mu_{F,t}(\mathbf{x},\mathbf{z})$ for BILBO and TrustedRand, and $\arg\max_{\mathbf{x}\in\mathcal{X}}\mu_{F,t}(\mathbf{x})$ for Nested. Additional implementation details are in Appendix D.2.

Results are averaged over 5 runs, and we compare the performance by plotting the instantaneous regret over number of queries with 95% confidence intervals. The instantaneous regret in this section is calculated as the sum of each function's instantaneous regret ($\sum_{h\in\mathcal{F}} r_{h,t}$) , to provide intuitive comparison across different methods. Initial observations are included in the number of queries, resulting in a slight gap in the regret plots, before the estimation of optimal points begin.

### 4.1 SYNTHETIC PROBLEMS

The synthetic problems were selected to cover a variety of scenarios, including conflicting interactions, convex or multimodal functions, and active constraints.

**BraninHoo+GoldsteinPrice** has the Branin-Hoo function as upper-level objective $F$ and the Goldstein-Price function as the lower level objective $f$ (Picheny et al., 2013). Both functions are non-convex and multimodal. The dimensions $d_\mathcal{X}$ and $d_\mathcal{Z}$ are both 1, which facilitates visualization of the models and queries. Both dimensions were discretized into 100 points. The Branin-Hoo function has 3 optimal points, but there is only 1 optimal bilevel solution when constrained by lower-level Goldstein-Price optimal solutions.

BILBO outperforms the other two methods by a substantial margin, as seen in Figure 1a, where it converges to the optimal bilevel solution within 150 queries. Nested BO and TrustedRand converge equally slowly. For Nested BO, the predicted lower-level solutions may be sub-optimal because the lower-level solver cannot handle multimodal functions and noisy observations effectively. For TrustedRand, random queries might have led to uninformative points being sampled, reducing sample efficiency. The challenging multimodal characteristic of the functions also means that sampling in informative areas is integral for this problem, which BILBO successfully manages to do.

Figures 1b and 1c show the upper- and lower-level objective function respectively, with the optimal lower-level solutions (yellow dots) plotted. BILBO converged to the optimal solution, so the optimal bilevel point (red dot) and predicted optimal point (green cross) are plotted at the same location. Using BILBO, the surrogate models effectively captured the overall landscape of both functions, as shown in Figures 1d and 1e, where the predicted lower-level solutions (yellow crosses) are very similar to the lower-level optimal solutions, especially in regions where upper-level objective $F$ is close to the optimal. The query points chosen by BILBO over iterations on the upper- and lower-level objective function are shown in Figures 1f and 1g respectively, with darker colors indicating points sampled in earlier iterations. We observed that the queries mostly clustered around two probable optimal solutions, and BILBO sampled the objective functions around the top-left region until it was satisfied that it is not an optimal solution, and converged on the actual optimal solution, demonstrating the effectiveness of BILBO.

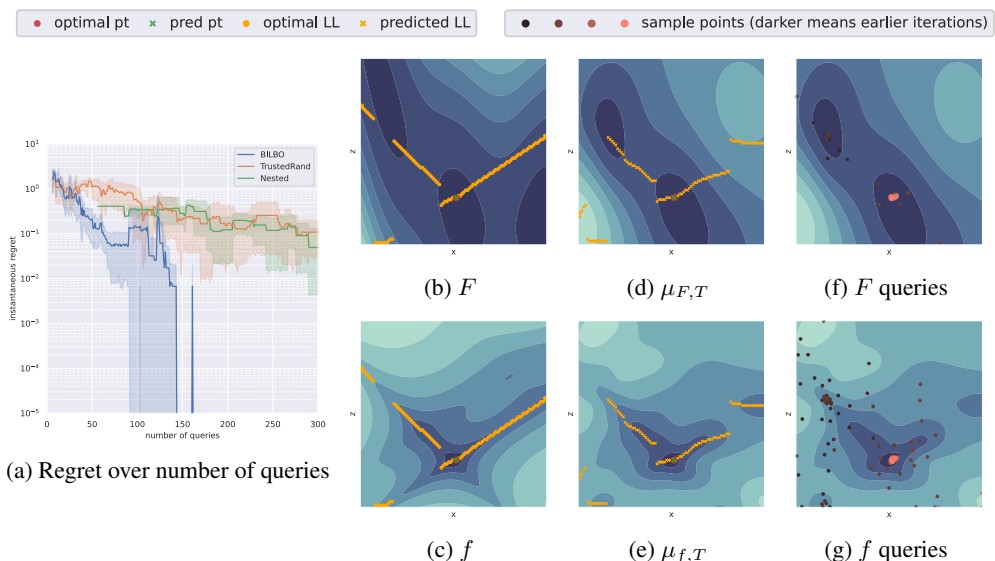

Figure 1: Results on BraninHoo+GoldsteinPrice experiment. (a) Comparison. (b) Upper-level objective, Branin-Hoo. (c) Lower-level objective, Goldstein-Price. (d-g) are BILBO outputs.

**SMD2**, **SMD6**, and **SMD12** are adapted from the SMD suite of test problems for bilevel optimization (Sinha et al., 2014). Details of implementation are in Appendix D.3. The input dimension of the test problems is set to 5, with $d_\mathcal{X}$ being 2 and $d_\mathcal{Z}$ being 3. The difficulty increases in the order of SMD2, SMD6, SMD12. SMD2 has convex functions and conflicting interactions, where improving the lower-level estimate worsens the upper-level objective value. This requires the algorithm to predict lower-level optimal solutions accurately in order to obtain the optimal bilevel solution. SMD6 also has convex functions and conflicting interactions, but with multiple lower-level optimal solutions at each upper-level point (i.e., a convex valley). An algorithm must concurrently estimate multiple lower-level optimal solutions and identify the point corresponding to the optimal upper-level objective. Finally, SMD12 is the most challenging problem from the SMD suite, where both levels have 3 active constraints, indicating that the optimal solution is on the boundary of the constraints. There are also multiple optimal solutions at the lower level.

Results of the SMD experiments are shown in Figure 2. For SMD2, BILBO outperforms both TrustedRand and Nested BO. While TrustedRand's regret decreased quickly at the start, its rate of decrease diminishes over time, likely because random queries are initially informative but become less effective as the process continues. For SMD6, BILBO has the smallest regret after around 250 steps. Nested BO is unable to handle multiple lower-level optimal solutions, as it only selects one lower-level optimal solution for each upper-level point. In comparison, the trusted sets allow multiple optimal lower-level estimates for both BILBO and TrustedRand. For SMD12, BILBO converges faster than TrustedRand. With 8 functions in the SMD12 problem, the decoupled setting becomes more crucial for sample efficiency. The faster convergence of BILBO demonstrates the effectiveness of its function query strategy in selecting more informative functions to query. The presence of active constraints did not appear to pose any difficulties for BILBO.

## 4.2 Real-world problems

**Energy.** We simulated a real-world problem in energy markets, where energy providers bid to supply an amount of electricity to consumers at the upper level to maximize their profits over three time periods. At the lower level, they optimize their operations by considering costs, demand responses to prices, and their ability to meet changing demands over time. There are 2 upper-level variables: price and quantity of electricity to bid, and 2 lower-level variables: the ramp limit for one power plant and the maximum power output at each period for another power plant. The lower-level variables adjust the characteristics of the power plants, affecting the overall optimal dispatch of electricity.

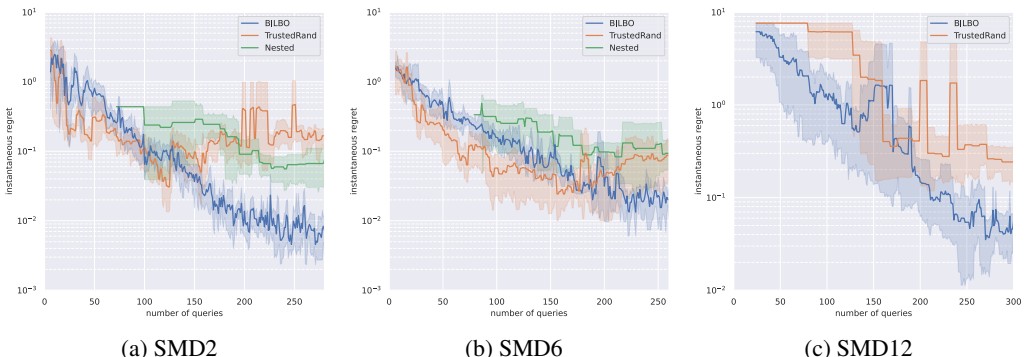

(a) SMD2        (b) SMD6        (c) SMD12

Figure 2: Instantaneous regrets over number of queries, averaged over 5 runs, for SMD2, SMD6 and SMD12 experiments.

The dispatching of three power plants was simulated using PyPSA, a Python library for Power Systems Analysis (Brown et al., 2018), and we formulated a lower-level objective function based on the simulation outputs. Ideally, we seek the lowest cost combination of electricity generation, incorporating penalties to reduce wear and tear or other auxiliary concerns. More details in Appendix D.4.

Results in Figure 3 show that BILBO outperforms the other two methods, with its regret over queries decreasing the fastest. We analyse the surrogate models learned from one run of BILBO in Figures 3b to 3e. Figures 3b and 3d, respectively, show the upper-level objective $F$ at the lower-level optimal solutions and the estimated upper-level objective $\mu_{F,T}$ at estimated lower-level optimal solutions. We observed that $\mu_{F,T}$ approximates $F$ well, especially at regions with high $F$ values, and had correctly predicted the upper-level optimal solution. Figures 3c and 3e, respectively, show the lower-level objective $f$ and the estimated lower-level objective $\mu_{f,T}$ at the optimal upper-level variable. At this upper-level point, $\mu_{f,T}$ captures the general trend of $f$, where points on the right of the image are more optimal. However, the optimal lower-level solution is at a boundary with high discontinuity. The surrogate model was unable to model this large step and thus predicted a sub-optimal lower-level solution, resulting in an empirical asymptotic regret bound seen in Figure 3a, compounded by noisy observations. This may be mitigated by adding a constraint function to represent the discontinuity in the lower-level objective function, and BILBO has shown the capability to handle active constraints effectively in previous synthetic experiments.

**Chemical.** Chemical processes in industries such as pharmaceuticals, petrochemicals, and food production often involve multiple stages, each requiring parameter optimization. Bilevel optimization simplifies this by dividing the overall process into smaller, more manageable problems, while still accounting for the interactions between different stages. We used COCO simulator to simulate carbonylation of Di-Methyl Ether (DME) to Methyl Acetate, adapted from the flowsheet provided by ChemSep. The upper-level problem focuses on maximizing the yield of Methyl Acetate at 99.9% purity through a distillation column, which takes in a reaction mixture comprising Methyl Acetate, unreacted DME, and by-products. This optimization depends on the outputs from the lower-level problem, which involves the carbonylation of DME to produce Methyl Acetate in a reactor. Additionally, an upper-level constraint is included to ensure the process is feasible by requiring a suitable temperature range, where chemicals are in their correct states. There is 1 upper-level variable: the number of levels in the distillation column,

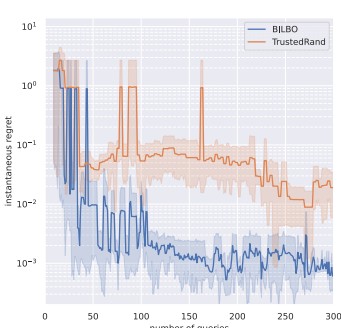

Figure 4: Results of chemical process experiment. Regret over queries.

and 3 lower-level variables: temperature of the reactor, number of heating tubes, and the diameter of heating tubes. More details are in Appendix D.5. Results are in Figure 4, where we can see that BILBO converges well, indicating the potential efficiency and effectiveness of BILBO in optimizing complex industrial operations.

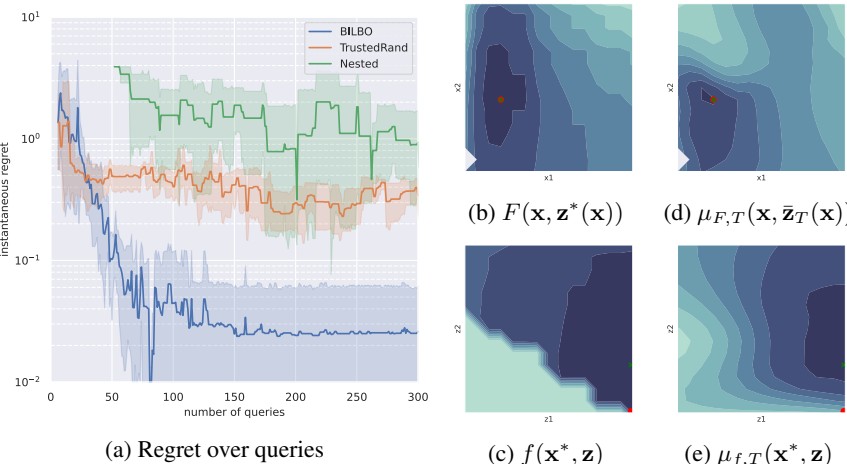

(b) $F(\mathbf{x}, \mathbf{z}^*(\mathbf{x}))$    (d) $\mu_{F,T}(\mathbf{x}, \bar{\mathbf{z}}_T(\mathbf{x}))$

(a) Regret over queries

(c) $f(\mathbf{x}^*, \mathbf{z})$    (e) $\mu_{f,T}(\mathbf{x}^*, \mathbf{z})$

Figure 3: Results of the energy experiment. (a) Comparison of regrets over 5 runs. (b) Upper-level objective at optimal lower-level solutions. (c) Lower-level objective at optimal upper-level variable. (d) Estimated upper-level objective at estimated lower-level solutions (e) Estimated lower-level objective at optimal upper-level variable. (d) and (e) are estimated by GPs from a BILBO run. For (b-d), we plot the optimal bilevel solution (red dot) and estimated bilevel solution (green cross).

## 5 FUTURE WORK

We have shown theoretically and empirically that BILBO is a regret-bounded, sample efficient algorithm for noisy, constrained, and derivative-free bilevel optimization. A key direction for future work is improving scalability to high-dimensional spaces, which is a common challenge in BO.

We currently model upper-level objective $F(\mathbf{x}, \mathbf{z})$ for $(\mathbf{x}, \mathbf{z}) \in \mathcal{X} \times \mathcal{Z}$, but this can be memory inefficient as many lower-level variables $\mathbf{z}$ are suboptimal and irrelevant. A more efficient approach could involve directly modeling $F(\mathbf{x}, \mathbf{z}^*(\mathbf{x}))$ for $\mathbf{x} \in \mathcal{X}$, reducing the dimension of the surrogate model from $d_{\mathcal{X}} \times d_{\mathcal{Z}}$ to $d_{\mathcal{X}}$. This poses another challenge: incorporating the uncertainty associated with the optimality of the lower-level solution into the uncertainty of the upper-level objective value.

Adaptive discretization (Shekhar & Javidi, 2017) may also reduce computational complexity by reducing the effective dimension of the explored space. Discretization strategies could be integrated with trusted sets, for example concentrating the discretizations to within the trusted sets, while taking into account constraint and objective estimates of both upper- and lower-level problems. Approximate surrogate models (Calandriello et al., 2019) offer another possible direction for scaling to high-dimensional functions while preserving confidence bound estimates. The theoretical work presented in this paper could be extended to selected approximate surrogate models.

The representation of trusted sets will need to scale effectively to higher dimensions as well. Possible approaches could be via sampling strategies like Latin Hypercube Sampling (McKay et al., 2000) for efficient point representation in high-dimensional spaces or using hyperrectangles to represent the trusted set efficiently (Eriksson et al., 2019).

## 6 CONCLUSION

We introduced BILBO, a novel bilevel Bayesian optimization algorithm that optimizes functions on the upper- and lower-levels simultaneously, using trusted sets to reduce the search space for querying. The trusted sets are constructed using confidence bounds, from which we derived instantaneous regret bounds for points in these sets, and lead to a sublinear cumulative regret bound in the decoupled setting. Our experiments also show that BILBO outperforms other bilevel optimization baselines, especially in problems with many non-convex functions. BILBO is a significant step towards a more general bilevel solver, which will enable applications to complex real-world bilevel problems involving blackbox functions.

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

## A    TABLE OF NOTATIONS

**Bilevel definitions**

| **Upper-level** | | **Lower-level** | |
|---|---|---|---|
| $\mathbf{x}$ | Upper-level variable | $\mathbf{z}$ | Lower-level variable |
| $\mathcal{X}$ | Domain of $\mathbf{x}$ | $\mathcal{Z}$ | Domain of $\mathbf{z}$ |
| $d_{\mathcal{X}}$ | Dimension of $\mathbf{x}$ | $d_{\mathcal{Z}}$ | Dimension of $\mathbf{z}$ |
| $F$ | Upper-level objective function | $f$ | Lower-level objective function |
| $\mathcal{C}_{\text{up}}$ | Set of upper-level constraint functions | $\mathcal{C}_{\text{lo}}$ | Set of lower-level constraint functions |
| $\mathbf{x}_t$ | Selected upper-level variable to query at time $t$ | $\mathbf{z}_t$ | Selected lower-level variable to query at time $t$ |
| $\hat{\mathbf{x}}_T$ | Estimated optimal upper-level variable at time $T$ | $\hat{\mathbf{z}}_t$ | Estimated optimal lower-level variable at time $T$ |

| | |
|---|---|
| $\mathcal{F}$ | Set of functions in a bilevel problem $\{F, f\} \cup \mathcal{C}_{\text{up}} \cup \mathcal{C}_{\text{lo}}$ |
| $h$ | Arbitrary function in $\mathcal{F}$ |
| $\mu_{h,t}(\mathbf{x}, \mathbf{z})$ | GP posterior mean at $(\mathbf{x}, \mathbf{z})$ for function $h$ at time $t$ |
| $\sigma_{h,t}(\mathbf{x}, \mathbf{z})$ | GP posterior standard deviation at $(\mathbf{x}, \mathbf{z})$ for function $h$ at time $t$ |
| $r_h(\mathbf{x}, \mathbf{z})$ | Instantaneous regret of function $h$ at $(\mathbf{x}, \mathbf{z})$ |
| $r_t$ | Instantaneous bilevel regret at time $t$ on query point $(\mathbf{x}_t, \mathbf{z}_t)$ |
| $R_T$ | Cumulative regret at time $T$ |
| $r_T$ | Simple bilevel regret at time $T$ based on $(\hat{\mathbf{x}}_t, \hat{\mathbf{z}}_t)$ |

**BILBO notations**

| | |
|---|---|
| $u_{h,t}(\mathbf{x}, \mathbf{z})$ | Upper confidence bound of function $h$ at $(\mathbf{x}, \mathbf{z})$ (Defn. 3.1) |
| $l_{h,t}(\mathbf{x}, \mathbf{z})$ | Lower confidence bound of function $h$ at $(\mathbf{x}, \mathbf{z})$ (Defn. 3.1) |
| $\mathcal{S}_t^+$ | Trusted set of feasible solutions (Defn. 3.3) |
| $\mathcal{S}_{\text{lo},t}^+$ | Trusted set of feasible solutions w.r.t. only lower-level constraints (Defn. 3.5) |
| $\mathcal{P}_t^+$ | Trusted set of optimal lower-level solutions (Defn. 3.5) |
| $\bar{\mathbf{z}}_t(\mathbf{x})$ | Estimated optimal lower-level solution at $\mathbf{x}$ at timestep $t$ (Defn. 3.5) |
| $h_t$ | Selected function query (Defn. 3.7) |
| $\bar{r}_{h,t}$ | Estimated regret for function $h$ (Defn. 3.7) |

## B    MORE PRELIMINARIES DETAILS

### B.1    CLOSED-FORM POSTERIORS OF GAUSSIAN PROCESSES

For a GP defined as $\mathcal{GP}_h(m_h(\mathbf{xz}), k_h(\mathbf{xz}, \mathbf{xz}'))$ for a function $h$. The closed-form posterior mean is $\mu_{h,t-1}(\mathbf{xz}) \triangleq m_h(\mathbf{xz}) + \mathbf{k}_{h,t-1}(\mathbf{xz})^\top (\mathbf{K}_{h,t-1} + \sigma^2 \mathbf{I})^{-1}(\mathbf{y}_{h,t-1} - \mathbf{m}_{h,t-1})$ and variance $\sigma_{h,t-1}^2(\mathbf{xz}) \triangleq k_h(\mathbf{xz}, \mathbf{xz}) - \mathbf{k}_{h,t-1}^\top (\mathbf{K}_{h,t-1} + \sigma^2 \mathbf{I})^{-1} \mathbf{k}_{h,t-1}^{-1}$ where $\mathbf{m}_{h,t-1} \triangleq [m_h(\mathbf{x}, \mathbf{z})]_{\mathbf{xz} \in \mathbf{xz}_{:t-1}}$, $\mathbf{k}_{h,t-1}(\mathbf{xz}) \triangleq [k_h(\mathbf{xz}, \mathbf{xz}')]_{\mathbf{xz}' \in \mathbf{xz}_{:t-1}}$, and $\mathbf{K}_{h,t-1} \triangleq [k_h(\mathbf{xz}, \mathbf{xz}')]_{\mathbf{xz}, \mathbf{xz}' \in \mathbf{xz}_{:t-1}}$.

## B.2 MAXIMUM INFORMATION GAIN

Maximum information gain on a function $h$ from Vakili et al. (2021), where $d \triangleq d_{\mathcal{X}} + d_{\mathcal{Z}}$:

- Squared Exponential kernel: $\mathcal{O}(\log^{d+1}(T))$
- Matérn kernels with $\nu > \frac{1}{2}$: $\mathcal{O}(T^{\frac{d}{2\nu+d}} \log^{\frac{2\nu}{2\nu+d}}(T))$

# C  PROOFS

## C.1  PROOF OF LEMMA 3.4

*Proof.* $\forall c \in \mathcal{C}_{\mathrm{up}} \cup \mathcal{C}_{\mathrm{lo}}, (\mathbf{x}, \mathbf{z}) \in \mathcal{S}_t^+$,

$$
\begin{aligned}
r_{c,t}(\mathbf{x}, \mathbf{z}) &\triangleq \max(0, -c(\mathbf{x}, \mathbf{z})) && \text{From Equation 2.7} \\
&\leq \max(0, -l_{c,t}(\mathbf{x}, \mathbf{z})) && \text{from Corollary 3.2} \\
&\leq \max(0, u_{c,t}(\mathbf{x}, \mathbf{z}) - l_{c,t}(\mathbf{x}, \mathbf{z})) && \text{from } (\mathbf{x}, \mathbf{z}) \in \mathcal{S}_t^+ \\
&\leq 2\beta_t^{1/2} \sigma_{c,t-1}(\mathbf{x}, \mathbf{z}). && \text{from Definition 3.1}
\end{aligned}
$$

$\square$

## C.2  PROOF OF LEMMA 3.6

**Lemma C.1.** $\forall \mathbf{x} \in \{\mathbf{x} \mid (\mathbf{x}, \mathbf{z}) \in \mathcal{P}_t^+\}$,

$$u_{f,t}(\mathbf{x}, \bar{\mathbf{z}}_t(\mathbf{x})) \geq u_{f,t}(\mathbf{x}, \mathbf{z}^*(\mathbf{x})), \tag{C.1}$$

*where $\bar{\mathbf{z}}_t(\mathbf{x}) \triangleq \arg\max_{\mathbf{z} \in \mathcal{S}_{lo,t}^+(\mathbf{x})} u_{f,t}(\mathbf{x}, \mathbf{z})$ is the estimated optimal lower-level solution at $\mathbf{x}$, and $\mathbf{z}^*(\mathbf{x})$ is the actual optimal lower-level solution at $\mathbf{x}$.*

*Proof.* By definition of $\bar{\mathbf{z}}_t(\mathbf{x})$, $\forall (\mathbf{x}, \mathbf{z}) \in \mathcal{S}_{\mathrm{lo},t}^+, u_{f,t}(\mathbf{x}, \bar{\mathbf{z}}_t(\mathbf{x})) \geq u_{f,t}(\mathbf{x}, \mathbf{z})$.

Let $\mathcal{S}_{\mathrm{lo}} \triangleq \{(\mathbf{x}, \mathbf{z}) \mid c(\mathbf{x}, \mathbf{z}) \geq 0 \ \forall c \in \mathcal{C}_{\mathrm{lo}}\}$ be the unknown set of feasible solutions w.r.t lower-level constraints. Then, $(\mathbf{x}, \mathbf{z}^*(\mathbf{x})) \in \mathcal{S}_{\mathrm{lo},t}^+$, because $(\mathbf{x}, \mathbf{z}^*(\mathbf{x})) \in \mathcal{S}_{\mathrm{lo}}$ by definition and $\mathcal{S}_{\mathrm{lo}} \subseteq \mathcal{S}_{\mathrm{lo},t}^+$ from Corollary 3.2.

Finally, by Definition 3.5 of $\mathcal{P}_t^+$, $\mathcal{P}_t^+ \subseteq \mathcal{S}_{\mathrm{lo},t}^+$.

$\square$

**Main proof** for instantaneous regret bound on $f$ in Lemma 3.6.

*Proof.* $\forall (\mathbf{x}, \mathbf{z}) \in \mathcal{P}_t^+$,

$$
\begin{aligned}
r_{f,t}(\mathbf{x}, \mathbf{z}) &= f(\mathbf{x}, \mathbf{z}^*(\mathbf{x})) - f(\mathbf{x}, \mathbf{z}) && \text{From Equation 2.6} \\
&\leq u_{f,t}(\mathbf{x}, \mathbf{z}^*(\mathbf{x})) - l_{f,t}(\mathbf{x}, \mathbf{z}) && \text{from Corollary 3.2} \\
&\leq u_{f,t}(\mathbf{x}, \bar{\mathbf{z}}_t(\mathbf{x})) - l_{f,t}(\mathbf{x}, \mathbf{z}). && \text{from Lemma C.1}
\end{aligned}
$$

For $\mathbf{z} = \bar{\mathbf{z}}_t(\mathbf{x})$,

$$
\begin{aligned}
r_{f,t}(\mathbf{x}, \mathbf{z}) &\leq u_{f,t}(\mathbf{x}, \bar{\mathbf{z}}_t(\mathbf{x})) - l_{f,t}(\mathbf{x}, \bar{\mathbf{z}}_t(\mathbf{x})) \\
&= 2\beta_t^{1/2} \sigma_{f,t-1}(\mathbf{x}, \bar{\mathbf{z}}_t(\mathbf{x})) && \text{from Definition 3.1}
\end{aligned}
$$

and for $\mathbf{z} \neq \bar{\mathbf{z}}_t(\mathbf{x})$,

$$
\begin{aligned}
r_{f,t}(\mathbf{x}, \mathbf{z}) &\leq u_{f,t}(\mathbf{x}, \bar{\mathbf{z}}_t(\mathbf{x})) - l_{f,t}(\mathbf{x}, \mathbf{z}) \\
&\leq u_{f,t}(\mathbf{x}, \bar{\mathbf{z}}_t(\mathbf{x})) - u_{f,t}(\mathbf{x}, \mathbf{z}) + 2\beta_t^{1/2} \sigma_{f,t-1}(\mathbf{x}, \mathbf{z}) && \text{from Definition 3.1} \\
&\leq u_{f,t}(\mathbf{x}, \bar{\mathbf{z}}_t(\mathbf{x})) - l_{f,t}(\mathbf{x}, \bar{\mathbf{z}}_t(\mathbf{x})) + 2\beta_t^{1/2} \sigma_{f,t-1}(\mathbf{x}, \mathbf{z}) && \text{from } (\mathbf{x}, \mathbf{z}) \in \mathcal{P}_t^+ \\
&= 2\beta_t^{1/2} \sigma_{f,t-1}(\mathbf{x}, \bar{\mathbf{z}}_t(\mathbf{x})) + 2\beta_t^{1/2} \sigma_{f,t-1}(\mathbf{x}, \mathbf{z}). && \text{from Definition 3.1}
\end{aligned}
$$

Combining both cases, we get the instantaneous regret for lower-level objective function as

$$r_{f,t}(\mathbf{x}, \mathbf{z}) \leq \mathbb{1}_{\mathbf{z} \neq \bar{\mathbf{z}}_t(\mathbf{x})} 2\beta_t^{1/2} \sigma_{f,t-1}(\mathbf{x}, \bar{\mathbf{z}}_t(\mathbf{x})) + 2\beta_t^{1/2} \sigma_{f,t-1}(\mathbf{x}, \mathbf{z}).$$

$\square$

### C.3  PROOF OF LEMMA 3.8

**Lemma C.2.**
$$(\mathbf{x}^*, \mathbf{z}^*) \in \mathcal{S}_t^+ \cap \mathcal{P}_t^+,$$

*where $(\mathbf{x}^*, \mathbf{z}^*)$ is the optimal bilevel solution.*

*Proof.* Let the unknown feasible set be $\mathcal{S} \triangleq \{(\mathbf{x}, \mathbf{z}) \mid c(\mathbf{x}, \mathbf{z}) \geq 0 \quad \forall c \in \mathcal{C}_{\text{up}} \cup \mathcal{C}_{\text{lo}}\}$. Since $(\mathbf{x}^*, \mathbf{z}^*) \in \mathcal{S}$ by definition and $\mathcal{S} \subseteq \mathcal{S}_t^+$ by Corollary 3.2, we have $(\mathbf{x}^*, \mathbf{z}^*) \in \mathcal{S}_t^+$.

Let unknown feasible set w.r.t. lower-level constraints be $\mathcal{S}_{\text{lo}} \triangleq \{(\mathbf{x}, \mathbf{z}) \mid c(\mathbf{x}, \mathbf{z}) \geq 0 \quad \forall c \in \mathcal{C}_{\text{lo}}\}$. Similarly, we have $(\mathbf{x}^*, \mathbf{z}^*) \in \mathcal{S}_{\text{lo}} \subseteq \mathcal{S}_{\text{lo},t}^+$. Since $u_{f,t}(\mathbf{x}^*, \mathbf{z}^*) \geq f(\mathbf{x}^*, \mathbf{z}^*) \geq f(\mathbf{x}^*, \bar{\mathbf{z}}_t(\mathbf{x}^*)) \geq l_{f,t}(\mathbf{x}^*, \bar{\mathbf{z}}_t(\mathbf{x}^*))$, we have $(\mathbf{x}^*, \mathbf{z}^*) \in \mathcal{P}_t^+$.

$$(\mathbf{x}^*, \mathbf{z}^*) \in \mathcal{S}_t^+ \text{ and } (\mathbf{x}^*, \mathbf{z}^*) \in \mathcal{P}_t^+ \Rightarrow (\mathbf{x}^*, \mathbf{z}^*) \in \mathcal{S}_t^+ \cap \mathcal{P}_t^+$$

$\square$

**Lemma C.3.** *For some small $\delta > 0$, with probability at least $1 - \delta$, the instantaneous upper-level objective regret is upper bounded at the query point,*

$$r_F(\mathbf{x}_t, \mathbf{z}_t) \leq 2\beta_t^{1/2} \sigma_{F,t-1}(\mathbf{x}_t, \mathbf{z}_t).$$

*Proof.*

$$
\begin{aligned}
r_F(\mathbf{x}_t, \mathbf{z}_t) &\triangleq F(\mathbf{x}^*, \mathbf{z}^*) - F(\mathbf{x}_t, \mathbf{z}_t) && \text{From Equation 2.5} \\
&\leq u_{F,t}(\mathbf{x}^*, \mathbf{z}^*) - l_{F,t}(\mathbf{x}_t, \mathbf{z}_t) && \text{from Corollary 3.2} \\
&\leq \max_{(\mathbf{x}, \mathbf{z}) \in \mathcal{S}_t^+ \cap \mathcal{P}_t^+} u_{F,t}(\mathbf{x}, \mathbf{z}) - l_{F,t}(\mathbf{x}_t, \mathbf{z}_t) && \text{from Lemma C.2} \\
&= u_{F,t}(\mathbf{x}_t, \mathbf{z}_t) - l_{F,t}(\mathbf{x}_t, \mathbf{z}_t) && \text{from } \mathbf{x}_t, \mathbf{z}_t \triangleq \arg\max_{\mathcal{S}_t^+ \cap \mathcal{P}_t^+} u_{F,t} \\
&= 2\beta_t^{1/2} \sigma_{F,t-1}(\mathbf{x}_t, \mathbf{z}_t). && \text{From Definition 3.1}
\end{aligned}
$$

$\square$

**Lemma C.4.** *Given the estimated regret of the selected function query $h_t$ at the query point by Definition 3.7, the instantaneous regret $r_t$ is upper bounded,*

$$r_t \leq \bar{r}_{h_t,t}(\mathbf{x}_t, \mathbf{z}_t).$$

*Proof.* Given Definition 3.7, Lemma C.3, Lemma 3.4, and Lemma 3.6, $\forall h \in \mathcal{F}$, we can see that $\bar{r}_{h,t}(\mathbf{x}_t, \mathbf{z}_t) \geq r_h(\mathbf{x}_t, \mathbf{z}_t)$. Then,

$$
\begin{aligned}
r_t &\triangleq \max_{h \in \mathcal{F}} r_h(\mathbf{x}_t, \mathbf{z}_t) && \text{From Equation 2.4} \\
&\leq \max_{h \in \mathcal{F}} \bar{r}_{h,t}(\mathbf{x}_t, \mathbf{z}_t) \\
&= \bar{r}_{h_t,t}(\mathbf{x}_t, \mathbf{z}_t).
\end{aligned}
$$

$\square$

**Main proof** for instantaneous regret bound in Lemma 3.8

*Proof.* By Lemma C.4, if $h_t = f$,

$$
\begin{aligned}
r_t &\leq \bar{r}_{f,t}(\mathbf{x}_t, \mathbf{z}_t) \\
&= \mathbb{1}_{\mathbf{z}_t \neq \bar{\mathbf{z}}_t(\mathbf{x}_t)} 2\beta_t^{1/2} \sigma_{f,t-1}(\mathbf{x}_t, \bar{\mathbf{z}}_t(\mathbf{x}_t)) + 2\beta_t^{1/2}\sigma_{f,t-1}(\mathbf{x}_t, \mathbf{z}_t) \qquad \text{From Definition 3.7} \\
&\leq 4\beta_t^{1/2} \max(\sigma_{f,t-1}(\mathbf{x}_t, \bar{\mathbf{z}}_t(\mathbf{x}_t)), \sigma_{f,t-1}(\mathbf{x}_t, \mathbf{z}_t)) \\
&= 4\beta_t^{1/2}\sigma_{f,t-1}(\mathbf{x}_t, \mathbf{z}_t),
\end{aligned}
$$

where the last line holds because we reassign $\mathbf{z}_t \triangleq \bar{\mathbf{z}}_t(\mathbf{x}_t)$ if $\sigma_{f,t-1}(\mathbf{x}_t, \bar{\mathbf{z}}_t(\mathbf{x}_t)) \geq \sigma_{f,t-1}(\mathbf{x}_t, \mathbf{z}_t)$ as in Equation 3.8.

Else if $h_t \in \mathcal{F}/\{f\}$,

$$
\begin{aligned}
r_t &\leq \bar{r}_{h_t,t}(\mathbf{x}_t, \mathbf{z}_t) \\
&= 2\beta_t^{1/2}\sigma_{h_t,t-1}(\mathbf{x}_t, \mathbf{z}_t) \\
&\leq 4\beta_t^{1/2}\sigma_{h_t,t-1}(\mathbf{x}_t, \mathbf{z}_t).
\end{aligned}
$$

Combining, we obtain

$$
\begin{aligned}
r_t &\leq 4\beta_t^{1/2}\sigma_{h_t,t-1}(\mathbf{x}_t, \mathbf{z}_t) \\
&\leq 4\beta_t^{1/2} \max_{h \in \mathcal{F}} \sigma_{h,t-1}(\mathbf{x}_t, \mathbf{z}_t).
\end{aligned}
$$

$\square$

## C.4 PROOF OF THEOREM 3.9

*Proof.* From Lemma 3.8 and by Cauchy-Schwarz inequality, we derive the cumulative regret as

$$
\begin{aligned}
R_T^2 &\leq T \sum_{t=1}^{T} r_t^2 \\
&\leq T \sum_{t=1}^{T} 16\beta_t \max_{h \in \mathcal{F}} \sigma_{h,t-1}^2(\mathbf{x}_t, \mathbf{z}_t) \\
&\leq 4T\beta_T \sum_{h \in \mathcal{F}} \sum_{t \in T(h)} 4\sigma_{h,t-1}^2(\mathbf{x}_t, \mathbf{z}_t) \\
&\leq 4T\beta_T \sum_{h \in \mathcal{F}} C_h \gamma_{h,T(h)} \\
&\leq 4T\beta_T \sum_{h \in \mathcal{F}} C_h \gamma_{h,T} \\
&\leq 4T|\mathcal{F}|\beta_T \max_{h \in \mathcal{F}} C_h \gamma_{h,T},
\end{aligned}
$$

where $T(h)$ contains the timesteps where function $h$ was queried, so $\gamma_{T(h)} \leq \gamma_T$, and

$$
R_T \leq \sqrt{4T|\mathcal{F}|\beta_T \max_{h \in \mathcal{F}} C_h \gamma_{h,T}},
$$

where $C_h \triangleq 8/\log(1 + \sigma_h^{-2})$, and $\gamma_{h,T}$ is the maximum information gain from noisy observations of $h$ at $(\mathbf{x}_t, \mathbf{z}_t), \forall t \in [T]$. The proof methodology follows Srinivas et al. (2010). $\square$

| experiment | length scale prior | $d_{\mathcal{X}}$ | $d_{\mathcal{Z}}$ | discrete points per dimension |
|---|---|---|---|---|
| BraninHoo+GoldsteinPrice | 0.2 | 1 | 1 | 100 |
| SMD2 | 0.7 | 2 | 3 | 25 |
| SMD6 | 0.2 | 2 | 3 | 25 |
| SMD12 | 0.4 | 2 | 3 | 16 |
| Energy | 0.4 | 2 | 2 | 15 |
| Chemical | 0.8 | 1 | 3 | 10 |

Table 1: Experiment parameters

## C.5 PROOF OF LEMMA 3.10

*Proof.*

$$r_T \leq \min_{(\mathbf{x}_t, \mathbf{z}_t) \in \{(\mathbf{x}_{t'}, \mathbf{z}_{t'})\}_{t' \in [T]}} \max_{h \in \mathcal{F}} \bar{r}_{h,t}(\mathbf{x}_t, \mathbf{z}_t) \qquad \text{From Equation 3.9 and Lemma C.4}$$

$$\leq \frac{1}{T} \sum_{t=1}^{T} \max_{h \in \mathcal{F}} \bar{r}_{h,t}(\mathbf{x}_t, \mathbf{z}_t)$$

$$\leq \frac{1}{T} \sum_{t=1}^{T} 4\beta_t^{1/2} \max_{h \in \mathcal{F}} \sigma_{h,t-1}(\mathbf{x}_t, \mathbf{z}_t) \qquad \text{From Appendix C.3}$$

$$\leq \sqrt{4|\mathcal{F}|\beta_T \max_{h \in \mathcal{F}} C_h \gamma_{h,T} / T}. \qquad \text{From Appendix C.4}$$

$\square$

## D EXPERIMENT DETAILS

### D.1 BASELINE DETAILS

**TrustedRand** implements a vanilla variant of the trusted sets $\mathcal{S}_t^+$ and $\mathcal{P}_t^+$, where mean $\mu$ is used instead of upper confidence bound $u$. Query points are then randomly sampled from the trusted set variants.

**Nested** uses the sequential least squares programming (SLSQP) optimizer for lower-level optimization and BO with upper confidence bound acquisition function Srinivas et al. (2010) at the upper-level. The lower-level problem is solved to convergence at each upper level query point. Note that SLSQP can only work on continuous functions.

### D.2 IMPLEMENTATION DETAILS

GP with Matérn 5/2 kernel was used, and the GP hyperparameters were automatically tuned at each iteration using maximum likelihood estimation on the past observations. The hyperparameters include length scale and prior mean. The prior mean initialized to 0 for all experiments since the output is already normalized. The initial length scale and other parameters for each experiment are set according to Table 1. For SMD2, energy, and chemical experiment, we sampled from $\bar{\mathcal{P}}_t \triangleq \{(\mathbf{x}, \bar{\mathbf{z}}_t(\mathbf{x})) \ \forall \mathbf{x} \in \mathcal{X}\}$ instead of $\mathcal{P}_t^+$ as it was empirically found to be better.

### D.3 EDITS TO SMD2, SMD6, SMD12

The selected SMD problems were adapted so the input ranges from 0 to 1, and the outputs have a mean of 0 and standard deviation of 1, for parameters $p = 1, r = 1, q = 2$, while ensuring that their characteristics and optimal points remain the same. The upper- and lower-level objective functions of SMD each have 3 components. The following only records edits to the original SMD problems. Refer to Sinha et al. (2014) for the original SMD problems.

Let $\mathbf{x} = [\hat{x}_{u1}, \hat{x}_{u2}]$ and $\mathbf{z} = [\hat{x}_{l1}, \hat{x}_{l2}]$, $\mathbf{x}, \mathbf{z} \in [0, 1]^d$.

**SMD2**. To bound the output for the given domain, we set

$$F_3 \triangleq -\sum_{i=1}^{r} (x_{u2}^i)^2 - \sum_{i=1}^{r} (x_{u2}^i - \log(0.99 * x_{l2}^i + 0.01))^2,$$

$$f_3 \triangleq \sum_{i=1}^{r} x_{u2}^i - \log(0.99 * x_{l2}^i + 0.01)^2,$$

where $\hat{x}_{u1} \triangleq (x_{u1} + 1)/3$, $\hat{x}_{u2} \triangleq (x_{u2} + 5)/6$, $\hat{x}_{u1} \triangleq (x_{l1} + 1)/3$, and $\hat{x}_{l2} \triangleq x_{l2}/e$.

**SMD6**. The different functions have imbalanced ranges. To balance the different functions in $f$, we set

$$\hat{f}_1 \triangleq f_1/d$$
$$\hat{f}_2 \triangleq f_2/d^2$$
$$\hat{f}_3 \triangleq f_3/d,$$

where $d = 3$, and use $\hat{f} \triangleq \hat{f}_1 + \hat{f}_2 + \hat{f}_3$ as the lower-level objective function. $\hat{x}_b \triangleq (x_b + 1)/3$, for $x_b \in \{x_{u1}, x_{u2}, x_{l1}, x_{l2}\}$.

**SMD12**. To bound the outputs in the domain, we set

$$F_3 \triangleq \sum_{i=1}^{r} (x_{u2}^i - 2)^2 + \sum_{i=1}^{r} \tanh |x_{l2}^i| - \sum_{i=1}^{r} (x_{u2}^i - \tanh x_{l2}^i)^2$$

$$f_3 \triangleq \sum_{i=1}^{r} (x_{u2}^i - \tanh x_{l2}^i)^2$$

We also edited the first upper level constraint to $x_{u2}^i - \tanh x_{l2}^i \geq 1$, $\forall i \in \{1, ..., r\}$, so it becomes an active constraint. One of the lower level constraint was also edited to bound its output range: $x_{l1}^j - \sum_{i=1, i\neq j}^{q} (x_{l1}^i)^3 \geq 0 \ \forall j \in \{1, ..., q\}$. We normalize $\hat{x}_{u1} \triangleq (x_{u1} + 5)/15, \hat{x}_{u2} \triangleq (x_{u2} + 1)/2, \hat{x}_{l1} \triangleq (x_{l1} + 5)/15$, and $\hat{x}_{l2} \triangleq (x_{l2} + \pi/2)/\pi$.

After the following adaptations, we take the mean over input dimensions to ensure that function values do not increase with dimensions. Finally, we normalize the outputs.

### D.4 ENERGY MARKET

Let $\mathbf{x} \triangleq [\mathbf{x}_1, \mathbf{x}_2]$, where $\mathbf{x}_1$ denotes a price to bid and $\mathbf{x}_2$ denotes a quantity in MW to supply at bid price. $\mathbf{x}_1 \in (0.01, 0.5), \mathbf{x}_2 \in (200, 500)$. We simulate a network with 3 generators that has to fulfill an estimated demand schedule for 3 periods. The generators' parameters are given in Table 2, where $\mathbf{z} \triangleq [\mathbf{z}_1, \mathbf{z}_2]$ are the lower-level variables. $\mathbf{z}_1 \in (0.0, 0.2), \mathbf{z}_2 \in (0.5, 1.5)$. These two variables were selected as a proxy for auxiliary concerns such as efficiency and maintainence costs, on top of operational costs.

The lower-level objective function is denoted as

$$f(\mathbf{x}, \mathbf{z}) \triangleq -\text{cost}(\mathbf{x}, \mathbf{z}) - 2.5 * w_r(\mathbf{z}_1) - 1.5 * w_w(\mathbf{z}_2),$$

where $\text{cost}(\mathbf{x}, \mathbf{z})$ is the operational cost of producing $\mathbf{z}_2$MW of power, simulated by PyPSA. $w_r(\mathbf{z}_1) \triangleq \exp(5 * \mathbf{z}_1) - 1$ and $w_w(\mathbf{z}_2) \triangleq -(\log(-0.75 * \mathbf{z}_2 + 1.15) - (-0.75 * \mathbf{z}_2 + 1.15)) - 0.797$, where $w_r$ and $w_w$ are different nonlinear weighting functions applied to $\mathbf{z}$. If dispatch is not feasible at a point, we set the lower-level objective value with an arbitrary large negative number, and the upper-level objective value at 0.

The upper-level objective function measures profit as

$$F(\mathbf{x}, \mathbf{z}) \triangleq \mathbf{x}_1 * \mathbf{x}_2 * \text{df}(\mathbf{x}_t) - \text{cost}(\mathbf{x}, \mathbf{z}),$$

where $\text{df}(\mathbf{x}_t) \triangleq \min(1, \exp(-10\mathbf{x}_1 + 0.25))$ returns a factor that simulates the demand response of consumers. This implies a disincentive for providers to bid at high prices, because consumers might choose to reduce their electricity usage or look for alternative providers.

We discretized the input space into 15 at each dimension.

| type | nominal power | marginal cost | quadratic marginal cost | ramp limit | max p factor |
|------|---------------|---------------|-------------------------|------------|--------------|
| coal | 200 | 0.005 | 0.0005 | $\mathbf{z}_1$ | - |
| gas | 100 | 0.015 | 0.0005 | 0.5 | - |
| wind | 60 | 0.02 | 0.005 | - | $\mathbf{z}_2$ |

Table 2: Parameters input into PyPSA generator. 'max p factor' refer to 'p_max_pu', the maximum power at a snapshot given as a fraction of nominal power.

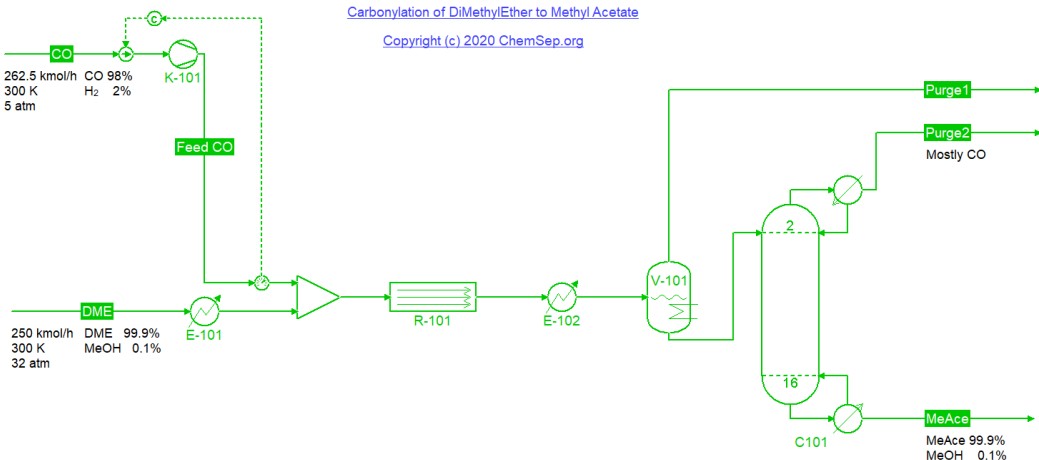

Figure 5: Flowsheet of chemical process. R-101 is the reactor, and C101 is the distillation column.

### D.5 CHEMICAL PROCESS

The flowsheet used is shown in Figure 5, where the output of reactor R101 contains a mix of Methyl Acetate, unreacted DME, and other by-products, and the distillation column C101 separates these products to obtain high purity Methyl Acetate. The flowsheet was adapted from ChemSep, where the recycle streams have been removed to simplify the process. CO and DME are fed in at a fixed flow rate and concentration for all experiments, as indicated in the figure. The distillation feed is always at level 2, and we fixed the output concentration of Methyl Acetate at 99.9%. Note that we can simulate the reactor R101 without the column C101.

The upper- and lower-level parameters to be optimized are defined in Table 3. We discretized the input space into 10 at each dimension, and the variables are normalized to $[0, 1]$.

Let $\text{sim}_{\text{R101}}(\mathbf{x}, \mathbf{z})$ be the simulated mass flow of Methyl Acetate (kg/s) at the output of the reactor R101, and $\text{sim}_{\text{C101}}(\mathbf{x}, \mathbf{z})$ be the simulated mass flow of Methyl Acetate (kg/s) at the MeAce output of the column C101.

The lower-level objective function is denoted as

$$f(\mathbf{x}, \mathbf{z}) \triangleq \text{sim}_{\text{R101}}(\mathbf{x}, \mathbf{z}) - 1\text{e-}3 * \mathbf{z}_1^4,$$

where the second term is a penalty on higher temperatures to account for energy costs.

The upper-level objective function is then denoted as

$$f(\mathbf{x}, \mathbf{z}) \triangleq \text{sim}_{\text{C101}}(\mathbf{x}, \mathbf{z}) - 1\text{e-}4 * \mathbf{x}_0^4,$$

where the second term is a penalty on more levels in the distillation column as it is associated with higher costs. The higher costs could be due to maintenance, energy consumption or equipment cost.

### D.6 COMPUTATIONAL RESOURCES

The experiments in this paper were done on a computer with AMD Ryzen 7 5700X 8-Core Processor and 64 GB of RAM.

| name | min | max | normalized symbol |
|---|---|---|---|
| Number of levels in distillation column | 5 | 23 | $\mathbf{x}_0 \in [0, 1]$ |
| Temperature of reactor (K) | 455 | 500 | $\mathbf{z}_0 \in [0, 1]$ |
| Number of heating tubes in reactor | 600 | 1500 | $\mathbf{z}_1 \in [0, 1]$ |
| Diameter of heating tubes (m) | 0.02 | 0.065 | $\mathbf{z}_2 \in [0, 1]$ |

Table 3: Parameters of the chemical experiment.

