# OpenReview forum: "BILBO: BILevel Bayesian Optimization"
_ICLR.cc/2025/Conference — Submitted to ICLR 2025_

### Official Review · Reviewer_pSWd · 2024-11-03

**Soundness:** 3
**Presentation:** 3
**Contribution:** 3
**Rating:** 6
**Confidence:** 3

**Summary:**

This paper studies the black-box bilevel optimization problem based on the Gaussian process (GP) model.
The author proposes an algorithm that leverages the confidence set of the underlying objective function $f$ under the Bayesian assumption (i.e., $f$ is the sample path of GP.)
The proposed algorithm achieves $O(\sqrt{\gamma_T T})$ cumulative regret upper bound whose
instantaneous regrets are defined as the extension of existing work [1] for the formulation of the bilevel optimization. The numerical experiments are conducted, including two problems motivated by real-world problems.

[1] Nguyen, Quoc Phong, et al. "Optimistic Bayesian Optimization with Unknown Constraints." The Twelfth International Conference on Learning Representations. 2023.

**Strengths:**

- Overall, the writing of this paper is clear, well-structured, and easy to follow.
- Comprehensive proofs are provided, and the proofs seem correct, as far as I can see.
- As far as I know, the proposed algorithm is the first theoretically guaranteed approach for bilevel Bayesian optimization.
- The practical behavior of the algorithm is well-described in Figures 1 and 2, which will be helpful for practitioners.

**Weaknesses:**

My main concern is the lack of novelty. The confidence bound-based approach for
complex structured optimization problems have already been extensively studied
in the BO field (e.g., robust BO [1,2,3], constrained BO [4,5], and composite objectives [6, 7], etc.).
Specifically, it seems that the proposed algorithm can be constructed by combining the extended regret definition of [4] and the sampling for potential solutions based on a confidence set.  As far as I see, the non-trivial points of the proposed algorithm construction are the following:

1. The construction of $\mathcal{P}_t$ with $\overline{z}$ (Eq. 3.4, 3.5). Specifically, the natural construction of the potential solution set $\mathcal{P}\_t$ is based on maximum $\max l\_{f,t}(x, z)$; however, we cannot bound $r\_{f}$ with such construction.
2. Reassignment of $z_t$ based on Lemma 3.6.

I believe that the other parts of the algorithm construction and analysis are not novel for the reader who studies related BO fields (e.g., robust BO, constrained BO, etc.) since they only use the basic, well-known result of the existing algorithm and are naturally derived.
Furthermore, uncertainty-based input selection by fixing one input, such as the procedures of point 2, is also commonly leveraged to upper bound the instantaneous regret in the BO field (e.g., [2,3]).

For the above reasons, I slightly lean my score toward rejection.

[1] Bogunovic, Ilija, et al. "Adversarially robust optimization with Gaussian processes." Advances in neural information processing systems 31 (2018).

[2] Kirschner, Johannes, et al. "Distributionally robust Bayesian optimization." International Conference on Artificial Intelligence and Statistics. PMLR, 2020.

[3] Iwazaki, Shogo, Yu Inatsu, and Ichiro Takeuchi. "Mean-variance analysis in Bayesian optimization under uncertainty." International Conference on Artificial Intelligence and Statistics. PMLR, 2021.

[4] Nguyen, Quoc Phong, et al. "Optimistic Bayesian Optimization with Unknown Constraints." The Twelfth International Conference on Learning Representations. 2023.

[5] Xu, Wenjie, et al. "Constrained efficient global optimization of expensive black-box functions." International Conference on Machine Learning. PMLR, 2023.

[6] Li, Zihan, and Jonathan Scarlett. "Regret bounds for noise-free cascaded kernelized bandits." arXiv preprint arXiv:2211.05430 (2022).

[7] Xu, Wenjie, et al. "Bayesian optimization of expensive nested grey-box functions." arXiv preprint arXiv:2306.05150 (2023).

(Minor)
- I believe that there is further room to enhance the paper's quality. For example:
    - The existing work [1] studies the particular case of the problem setting of this paper. The comparison with [1] will make the position and novelty of this paper clearer.
    - Finiteness assumptions of $\mathcal{X}$ and $\mathcal{Z}$ should be explicitly described in Section 2.
    - The references should be appropriately capitalized (e.g., bayesian -> Bayesian).
    - The notation $|\mathcal{X}|$ usually denotes the cardinality of the set, not the dimension. I recommend using another notation.
    - Definition of the maximum information gain and its explicit upper bound for several commonly used kernels (SE or Mat\'ern) is beneficial for the reader who are not familiar with theory.
    - The vector graphics figures (such as .pdf, .svg) are favorable.
    - The result of Section.4 seems yet unstable in $5$ trials.
- (typo) Corollary 3.2: $[u\_{h,t}(x, z), l\_{h,t}(x, z)]$ -> $[l\_{h,t}(x, z), u\_{h,t}(x, z)]$
- (typo) Line 107: The definition of the posterior mean misses the prior mean $m_h(\cdot)$.
- (typo) Line 108: \sigma^2 I ->  \sigma^2 \bm{I}

**Questions:**

- Are there any problems with extending the proposed method for an infinite input set? I believe that we can extend it by relying on the discretization arguments (as in [1]) under the four-times differentiability assumption of the kernel and the compactness of the input set.
- Why does the author focus on the cumulative regret in Theorem 3.9? As with the existing works (e.g.,[2,3])
the upper bound of the simple regret (or the stopping time upper bound for finding $\epsilon$-accurate solution) seems to be derived based on the pessimistic estimated solution. I believe that simple regret is a more suitable performance measure from the optimization perspective.

[1] Srinivas, Niranjan, et al. "Gaussian process optimization in the bandit setting: No regret and experimental design." arXiv preprint arXiv:0912.3995 (2009).

[2] Bogunovic, Ilija, et al. "Adversarially robust optimization with Gaussian processes." Advances in neural information processing systems 31 (2018).

[3] Kirschner, Johannes, et al. "Distributionally robust Bayesian optimization." International Conference on Artificial Intelligence and Statistics. PMLR, 2020.

---

> ### Author Response · Authors · 2024-11-17
> **Official Comment by Authors (1/2)**
>
> We sincerely appreciate your detailed feedback and we have incorporated all minor edits suggested into the updated paper. Thank you for recognizing our proposed algorithm as the first theoretically guaranteed approach for bilevel Bayesian optimization, our comprehensive proofs, the clarity of our paper, and the well-described practical behavior in our figures.
>
> We would like to address the following concerns:
>
> >The confidence bound-based approach for complex structured optimization problems have already been extensively studied in the BO field (e.g., robust BO [1,2,3], constrained BO [4,5], and composite objectives [6, 7], etc.). Specifically, it seems that the proposed algorithm can be constructed by combining the extended regret definition of [4] and the sampling for potential solutions based on a confidence set. As far as I see, the non-trivial points of the proposed algorithm construction are the following:
> >1. The construction of $\mathcal{P}_t$ with $\bar{\mathbf{z}}$ (Eq. 3.4, 3.5)...
> >2. Reassignment of $\mathbf{z}_t$ based on Lemma 3.6.
>
> >Furthermore, uncertainty-based input selection by fixing one input, such as the procedures of point 2, is also commonly leveraged to upper bound the instantaneous regret in the BO field (e.g., [2,3]).
>
> >[2] Kirschner, Johannes, et al. "Distributionally robust Bayesian optimization." International Conference on Artificial Intelligence and Statistics. PMLR, 2020.
>
> > [3] Iwazaki, Shogo, Yu Inatsu, and Ichiro Takeuchi. "Mean-variance analysis in Bayesian optimization under uncertainty." International Conference on Artificial Intelligence and Statistics. PMLR, 2021.
>
> > [4] Nguyen, Quoc Phong, et al. "Optimistic Bayesian Optimization with Unknown Constraints." The Twelfth International Conference on Learning Representations. 2023
>
>
> We appreciate your recognition of trusted set $\mathcal{P}^+_t$ and reassignment of $\mathbf{z}_t$ during function query selection as non-trivial components of our work. However, we would like to clarify that our proposed algorithm tackles challenges in bilevel Bayesian optimization that have not been studied in other BO field and is not a straightforward extension of [4].
>
> Firstly, in bilevel optimization, the upper-level optimization is constrained by the unknown set of optimal lower-level solutions. This poses unique challenges not seen in robust BO, constrained BO or composite objectives BO, where they only tackle single level problems. Bilevel optimization algorithms have to consider the optimality of the estimated lower-level solutions, as querying upper-level functions at suboptimal lower-level solutions is likely largely uninformative.
>
> The trusted set of optimal lower-level solutions $\mathcal{P}^+_t$ was carefully designed to bound lower-level objective regret $r_f$ for points in the set. The regret bound is important to ensure that points sampled from this set are useful for upper-level querying, without requiring separate global optimization of the lower-level problem. This is a novel approach to bilevel Bayesian optimization and significantly more sample efficient than existing bilevel Bayesian optimization works [8, 9] which require separate global optimization of the lower-level problem at every upper-level query point.
>
> Secondly, the function query selection strategy which involves a conditional reassignment of lower-level query point $\mathbf{z}_t$ is another signification contribution of our work. The conditional reassignment of $\mathbf{z}_t$ to $\bar{\mathbf{z}}_t$ for lower-level objective function query is designed to manage uncertainty of estimated lower-level solutions. The need for reassignment arises because we do not carry out global optimization of lower-level problems at any upper-level point. This reassignment is used to bound the instantaneous regret of lower-level objective at the query point, and is specific to the challenge of bilevel optimization being constrained by optimal lower-level solutions.
>
> It differs from [2, 3] which has an additional unknown/uncontrollable variable $\mathbf{w}$ represented by a probability distribution. We do not fix $\mathbf{z}_t$ or sample $\mathbf{z}_t$ from another distribution. Both $\mathbf{z}_t$ and $\bar{\mathbf{z}}_t$ are selected based on confidence bounds on the objective functions, and we aim to find the optimal bilevel solution $\mathbf{x}^*, \mathbf{z}^* \in \mathcal{X} \times \mathcal{Z}$ whereas the optimal solution in [2, 3] is $\mathbf{x}^*$ and obtained via an expectation over $\mathbf{w}$. Importantly, since the problem contexts are different, the proposed algorithms in [2, 3] cannot be used to solve our bilevel optimization problem. We would appreciate if you could clarify on perceived similarities of reassignment of $\mathbf{z}_t$ to [2, 3].

---

> > ### Author Response · Authors · 2024-11-17
> > **Official Comment by Authors (2/2)**
> >
> > We are also the first paper to conduct theoretical analysis and provide regret bounds for bilevel Bayesian optimization. Our regret bounds, especially on the lower-level objective regret, formally validates the challenge of bilevel problems to obtain optimal lower-level solutions.
> >
> > In addition, most existing BO approaches do not handle decoupled settings where only one function is selected for query at every step. We showed that our function query selection based on estimated regrets and conditional reassignment leads to an instantaneous regret bound on the query point, which is a contribution to the BO field for problems in decoupled settings.
> >
> > We hope these clarifies that we have made significant and novel contributions to address challenges unique to bilevel optimization, which have not been tackled in other related BO fields, as well as to the general BO field in decoupled settings.
> >
> > [8] Kieffer, E., Danoy, G., Bouvry, P., & Nagih, A. (2017). Bayesian optimization approach of general bi-level problems. In Proc. Genetic and Evolutionary Computation Conf. Companion.
> >
> > [9] Dogan, V., & Prestwich, S. (2023). Bilevel optimization by conditional Bayesian optimization. In Int. Conf. on Machine Learning, Optim., and Data Sci.
> >
> > >The existing work [1] studies the particular case of the problem setting of this paper. The comparison with [1] will make the position and novelty of this paper clearer.
> >
> > >[1] Bogunovic, Ilija, et al. "Adversarially robust optimization with Gaussian processes." Advances in neural information processing systems 31 (2018).
> >
> > Our problem setting differs from [1]. [1] studies adversarially robust GP optimization and extends to robustness against unknown parameters $\theta \in \Theta$, where the goal is to find $\mathbf{x}^*$ that solves $\max_{\mathbf{x}\in\mathcal{X}} \min_{\theta\in\Theta} f(\mathbf{x},\theta)$. In comparison, we are studying bilevel optimization which aims to find $(\mathbf{x}^*, \mathbf{z}^*)$ that solves $\max_{\mathbf{x} \in \mathcal{X}, \mathbf{z} \in \mathcal{P}(\mathbf{x})} F(\mathbf{x}, \mathbf{z})$ (simplified unconstrained problem), where $\mathcal{P}(\mathbf{x})$ is the set of unknown optimal lower-level solutions at upper-level $\mathbf{x}$. The lower-level problem has its own set of black-box functions, which requires separate observations and modeling of, in order to estimate the optimal lower-level solutions for the upper-level problem.
> >
> > Key differences:
> > 1. Robust optimization in [1] focuses on a single decision variable $\mathbf{x}$ influenced by some uncertainty or perturbation parameter $\theta$, while our problem of bilevel optimization involves jointly optimizing both $\mathbf{x}$ and $\mathbf{z}$, and considering hierarchical dependencies between upper- and lower-levels.
> > 2. The domain $\Theta$ is known for robust optimization, while the domain $\mathcal{P}$ of $\mathbf{z}$ at the upper-level problem comprises of unknown optimal solutions of the lower-level problem. In bilevel optimization, the challenge is in obtaining good estimates of lower-level solutions, for optimization of the upper-level problem.
> >
> > This challenge is unique to bilevel optimization, and as mentioned above, we proposed a novel sample efficient bilevel Bayesian optimization that estimates optimal lower-level solutions without repeated global optimization of the lower-level problem, unlike any existing bilevel solutions or Bayesian optimization variants.
> >
> > >Are there any problems with extending the proposed method for an infinite input set? I believe that we can extend it by relying on the discretization arguments (as in [1]) under the four-times differentiability assumption of the kernel and the compactness of the input set.
> >
> > An infinite input set is an interesting future research direction to explore, however there are non-trivial challenges to address. For example, finding the trusted set of optimal lower-level solution $\mathcal{P}^+_t$ is not possible as we have to solve a maximization problem (equation 3.5) for a possibly infinite set of input.
> >
> > >Why does the author focus on the cumulative regret in Theorem 3.9? ... I believe that simple regret is a more suitable performance measure from the optimization perspective.
> >
> > Thank you for your suggestion. We have added Lemma 3.10 in the updated paper to include a simple regret bound. Lemma 3.10 is derived from Theorem 3.9. For an estimator defined in Equation 3.9 of the updated paper (refer to updated paper, as latex formatting is not working for this equation), we obtain a simple regret of $r_T \leq \sqrt{4 |\mathcal{F}| \beta_T \max_{h \in \mathcal{F}} C_h \gamma_{{h, T}} / T}$.
> >
> > We hope these clarifications on the novelty of our work, especially on challenges unique to bilevel optimization not tackled by other BO variants, and derivation of a simple regret bound will improve your opinion of our work.

---

> > > ### Author Response · Authors · 2024-11-24
> > > **Seeking Your Valued Feedback on Our Response**
> > >
> > > We hope this message finds you well. We are following up on our recent response to your questions and reviews of our paper and would like to know whether our response adequately addressed all your concerns. Your feedback is crucial to us, so could you kindly take a moment to confirm if our rebuttal meets your expectations? We greatly appreciate the time and effort you have devoted to responding to us.

---

> > > > ### Comment · Reviewer_pSWd · 2024-11-24
> > > >
> > > > I would like to thank the author's detailed reply.
> > > >
> > > > > Our problem setting differs from [1]. [1] studies adversarially robust GP optimization and extends to robustness against unknown parameters $\theta \in \Theta$
> > > >
> > > > I agree with the difference between this paper and the robust BO formulation [1] that the author pointed out.
> > > > On the other hand, I believe that the problem setting of [1] in Chapter 4 can be reduced to the problem setting that this paper considers.
> > > > To my understanding, the existing robust BO problem setting [1] is a special
> > > > case of this paper's setting Eq. (2,1)-(2.3) when the lower
> > > > level objective function $f(x, z)$ is also defined based on the upper-level objective function ($f(x, z) := - F(x, z)$), and there is no-constraint functions.
> > > >
> > > > > In addition, most existing BO approaches do not handle decoupled settings where only one function is selected for query at every step.
> > > >
> > > > To my understanding, the adaptation for decoupled settings is derived
> > > > from the design of the regret Eq. (2.4) and uncertainty-based query point selection.
> > > > I think that other existing uncertainty-based methods can also adapt the decoupled setting if we use the regret that is defined based on the maximum of each function's regret (e.g., I believe that existing constraint BO [2] can easily adapt the decuple setting by considering the similar regret formulation as Eq. (2.4)).
> > > > If my understanding is wrong, I would like the author to point out my misunderstanding
> > > > and clarify any technical challenge that the author solves to adapt the decouple setting in a bi-level optimization setting.
> > > >
> > > >
> > > > > Definition of the maximum information gain and its explicit upper bound for several commonly...
> > > >
> > > > The revised version of the paper describes the maximum information gain upper bound
> > > > of Srinivas et al. (2010) in Appendix B.2; however, I recommend the author use a refined version of the upper bound of the maximum information gain in [3].
> > > >
> > > > [1] Bogunovic, Ilija, et al. "Adversarially robust optimization with Gaussian processes." Advances in neural information processing systems 31 (2018).
> > > >
> > > > [2] Xu, Wenjie, et al. "Constrained efficient global optimization of expensive black-box functions." International Conference on Machine Learning. PMLR, 2023.
> > > >
> > > > [3] Vakili, Sattar, Kia Khezeli, and Victor Picheny. "On information gain and regret bounds in Gaussian process bandits." International Conference on Artificial Intelligence and Statistics. PMLR, 2021.

---

> ### Author Response · Authors · 2024-11-25
>
> Thank you for your reply.
>
> >To my understanding, the existing robust BO problem setting [1] is a special case of this paper's setting Eq. (2,1)-(2.3) when the lower level objective function $f(x, z)$ is also defined based on the upper-level objective function ($f(x, z) := -F(x, z)$), and there is no-constraint functions.
>
> While it is technically possible to formulate the robust BO problem setting [1] as a special case of bilevel optimization as you have stated, this formulation is not particularly meaningful. Solving the lower-level problem can directly contribute to the upper-level solution since the objective functions have the same blackbox component. This removes a challenge of bilevel optimization where suboptimal lower-level solutions are largely uninformative for the upper-level optimization. Also, many bilevel problems such as on toll setting and energy market optimization cannot be framed as this special case, and thus cannot be solved by robust BO but can be solved by our bilevel Bayesian optimization. Hence, our work on bilevel Bayesian optimization is a significant contribution.
>
> >(e.g., I believe that existing constraint BO [2] can easily adapt the decouple setting by considering the similar regret formulation as Eq. (2.4)). If my understanding is wrong, I would like the author to point out my misunderstanding and clarify any technical challenge that the author solves to adapt the decouple setting in a bi-level optimization setting.
>
> Without the complications of the lower-level problem, constrained BO [2] can adapt the decoupled settings by considering the upper-level objective and constraint regret formulation as you have pointed out. Indeed, this has been done in [4].
>
> However, a key technical challenge for the decoupled setting in bilevel optimization is in reducing the lower-level objective regret at the query point. We found out that simply selecting function queries based on estimated regrets at the query point is insufficient to reduce lower-level objective regret, as the query point is selected based on the upper-level objective function. Only observing the lower-level objective at these query points will not reduce the uncertainty of estimated lower-level solutions, since these query points were sampled from the trusted set $\mathcal{P}^+_t$ and may not be the estimated optimal lower-level solutions.
>
> Thus, for the lower-level objective function query, we reassign $\mathbf{z}_t$ to be the estimated lower-level solution $\bar{\mathbf{z}}_t(\mathbf{x}_t)$ when $\sigma_{f,t-1}(\mathbf{x}_t, \bar{\mathbf{z}}_t(\mathbf{x}_t)) \geq \sigma_{f,t-1}(\mathbf{x}_t, \mathbf{z}_t)$ (as in Eq. (3.8), latex formatting not working well here). This is related to the presence of both uncertainty terms in the estimated lower-level objective regret (Def. (3.7)), and we showed that our conditional reassignment will lead to a query point that can lower the estimated lower-level objective regret more effectively and lead to an instantaneous regret bound on the query point (in proof of Lemma 3.8).
>
> [4] Nguyen, Quoc Phong, et al. "Optimistic Bayesian Optimization with Unknown Constraints." The Twelfth International Conference on Learning Representations. 2023.
>
> >however, I recommend the author use a refined version of the upper bound of the maximum information gain in [3].
>
> Thank you for your recommendation. We have revised the upper bound of the maximum information gain to that in [3] in the updated paper.
>
> We hope these additional clarifications, especially on the technical challenges of bilevel optimization in the decoupled setting and our methods to tackle them, will improve your opinion of our paper.

---

> > ### Comment · Reviewer_pSWd · 2024-11-27
> >
> > Thank you for your reply. I have updated my score.
> >
> > However, since numerous algorithms based on confidence bounds for complex optimization problems exist in existing research, I believe the presentation in the revised version should be revised to better clarify the non-trivial technical differences by comparing the existing approaches and the author's proposed algorithm.
> >
> > Additionally, the revised version should carefully address the overall quality of the manuscript, including the above-mentioned points. For instance, in the current version, the definition of $\beta_t$ in Definition 3.1, Theorem 3.9, and Lemma 3.10 is incorrect. Specifically, the argument of the logarithm should be the cardinalities of the sets, $|\mathcal{X}|$ and $|\mathcal{Z}|$, rather than the dimensions $d_{\mathcal{X}}$ and $d_{\mathcal{Z}}$.

---

> > > ### Author Response · Authors · 2024-11-27
> > > **Thank you for your valuable feedback and improved score**
> > >
> > > Thank you for your valuable feedback and improved score. We appreciate your constructive feedback and the opportunity to clarify aspects of our contributions.  We have included the discussions here into the updated paper. We have also corrected the $\beta_t$ definition, thank you for pointing it out.

---

### Official Review · Reviewer_xVUG · 2024-11-04

**Soundness:** 3
**Presentation:** 3
**Contribution:** 3
**Rating:** 6
**Confidence:** 3

**Summary:**

This paper introduces a new algorithm designed to solve bi-level constrained Bayesian optimization problems. It demonstrates that the algorithm achieves a sublinear regret bound and provides experimental results on both synthetic and real-world datasets.

**Strengths:**

1. The problem addressed in this work is significant, and the proposed algorithm introduces a novel approach. Most existing bi-level Bayesian optimization algorithms tackle the problem using a nested framework, where each upper-level query requires separately optimizing the lower-level problem to convergence. In contrast, the proposed algorithm optimizes both levels concurrently, greatly enhancing sample efficiency, as shown in the experiments.
2. An infeasibility declaration method is also included.
3. Additionally, the algorithm achieves a theoretical sublinear regret bound, extending the classic bound from single-level to bi-level problems.

**Weaknesses:**

This approach involves keeping track of a set of nearly optimal solutions for the lower-level problem. In low-dimensional cases, this can be managed through discretization. However, it likely becomes difficult to scale effectively to even moderately high-dimensional problems.

**Questions:**

Have the authors considered a hybrid setting, where some parts of the bi-level problem are black-box while other parts are white-box with explicit expressions? I guess such problems also arise widely in practice.

---

> ### Author Response · Authors · 2024-11-17
>
> Thank you for recognizing our novel approach to bilevel Bayesian optimization, the significance of the problem addressed, our infeasibility declaration, and theoretical sublinear regret bound. We appreciate your valuable review.
>
> We would like to address the following concerns and questions:
>
> >This approach involves keeping track of a set of nearly optimal solutions for the lower-level problem. In low-dimensional cases, this can be managed through discretization. However, it likely becomes difficult to scale effectively to even moderately high-dimensional problems.
>
> This is indeed a common challenge for existing works on Gaussian processes and Bayesian optimization which require maintenance of a set, where discretization is often used. The need to maintain a set of near-optimal lower-level solutions is integral to our approach as it enables joint optimization of upper- and lower-level problems and handles noisy observations of lower-level functions more effectively.
>
> As we also pointed out in our future work section, possible approaches to scale to higher dimensions could be via sampling strategies like Latin Hypercube Sampling [1] for efficient point representation in high-dimensional spaces or using hyperrectangles to represent the trusted set efficiently [2].
>
> [1] McKay, M. D., Beckman, R. J., & Conover, W. J. (2000). A comparison of three methods for selecting values of input variables in the analysis of output from a computer code. Technometrics, 42(1), 55-61.
>
> [2] Eriksson, D., Pearce, M., Gardner, J., Turner, R. D., & Poloczek, M. (2019). Scalable global optimization via local Bayesian optimization. Advances in neural information processing systems, 32.
>
> >Have the authors considered a hybrid setting, where some parts of the bi-level problem are black-box while other parts are white-box with explicit expressions? I guess such problems also arise widely in practice.
>
> White-box problems with explicit expressions are often inexpensive to compute. We can still apply our algorithm, evaluate white-box parts at every step and reduce the confidence interval to a point, since there will be no noisy observations.
>
> We hope our responses have addressed your concerns and improve your opinion of our work.

---

> > ### Author Response · Authors · 2024-11-24
> > **Seeking Your Valued Feedback on Our Response**
> >
> > We hope this message finds you well. We are following up on our recent response to your questions and reviews of our paper and would like to know whether our response adequately addressed all your concerns. Your feedback is crucial to us, so could you kindly take a moment to confirm if our rebuttal meets your expectations? We greatly appreciate the time and effort you have devoted to responding to us.

---

> ### Comment · Reviewer_xVUG · 2024-11-25
> **Thanks for the response**
>
> I would like to thank the authors for the detailed responses. They addressed my concerns adequately. And I am happy to maintain my rating.

---

> > ### Author Response · Authors · 2024-11-25
> > **Thank you for your valuable feedback**
> >
> > Thank you for your valuable feedback. We are glad to have addressed your concerns.

---

### Official Review · Reviewer_7vtx · 2024-11-04

**Soundness:** 3
**Presentation:** 3
**Contribution:** 3
**Rating:** 6
**Confidence:** 3

**Summary:**

This paper introduces BILBO, a novel bilevel Bayesian optimization framework that simultaneously optimizes functions at both upper and lower levels. The algorithm employs trusted sets based on confidence bounds at each level. The authors establish a sublinear cumulative regret bound in a decoupled setting and validate BILBO through experiments on simulated and real-world problems, using TrustedRand and Nested policies as baselines.

**Strengths:**

- The paper proposes a new algorithm, BILBO, specifically tailored for general bilevel optimization problems, where simultaneous optimization at both levels is essential.
- The authors derive and prove a sublinear cumulative regret bound for the decoupled setting, enhancing the theoretical understanding of bilevel optimization.
- Experiments are conducted on both synthetic and interesting real-world scenarios, demonstrating the practical applicability of BILBO.
- The writing is generally clear and well-structured, making the paper accessible to readers.

**Weaknesses:**

- While the algorithm is well-explained, the motivations for certain design choices, particularly the decoupled setting and the specific function query mechanisms ($\mathcal{F}$ and $h_t$), are not sufficiently discussed. A deeper exploration of these choices would help readers understand the broader implications and advantages of BILBO’s design.
- The paper’s discussion of related work is somewhat limited. A more detailed comparison with existing bilevel optimization methods, particularly those offering theoretical guarantees or practical performance benchmarks, would provide a clearer picture of BILBO’s relative strengths and weaknesses.
- The baselines used in the experiments—TrustedRandom and Nested—are not the strongest available. Including comparisons with state-of-the-art bilevel optimization algorithms, such as those from Kieffer et al. (2017) or Dogan & Prestwich (2023), would offer a more rigorous evaluation of BILBO’s performance.
- No code is provided alongside the paper. This omission hampers the reproducibility of the results and limits the ability of the research community to build upon or validate the findings presented.

**Questions:**

- Can you explain more about the motivations for studying the decoupled setting? And the reasons to consider $\mathcal{F}$ and function query $h_t$ is the algorithm design and regret definition? For example, can we only focus on the upper and lower-level functions in the regret definition?
- For regret bound in Theorem 3.9, the authors discussed the relationship to constrained Bayesian optimization in Nguyen et al. (2023), is there any existing bilevel BO regret result that can be compared with?
- In experiments, BILBO is compared with TrustedRandom and Nested. As described in Appendix C.1, the Nested baseline used a nested approach with SLSDP. But the related works mentioned, e.g.  Kieffer et al. (2017); Dogan & Prestwich (2023) are not included. Can you compare it with the associated works directly? If not, can you explain the reasons that BILBO cannot compared with the related work?
- In Figure 1a, can you explain why BILBO has a sudden drop at around 150 queries? And what is the regret of BILBO after 150 queries (cannot observe the line afterwards)? The uncertainty level looks huge as well, I am wondering what would happen if you increase the number of runs (or change random seeds/initialisations, etc).
- In experimental results, we observe Nested is generally similar to or worse than TrustedRand (which is only a random sampling), can you explain the possible reason? Does it suggest Nested is not a strong baseline (as it is from a previous work)?

---

> ### Author Response · Authors · 2024-11-17
> **Official Comment by Authors (1/2)**
>
> We appreciate your valuable review and thank you for recognizing BILBO’s simultaneous optimization of both levels, our theoretical analysis, our experiments demonstrating practical applicability of BILBO, and our clear and well-structured writing.
>
> We would like to address the following concerns and questions:
>
> >While the algorithm is well-explained, the motivations for certain design choices, particularly the decoupled setting and the specific function query mechanisms ($\mathcal{F}$ and $h_t$), are not sufficiently discussed.
>
> >Can you explain more about the motivations for studying the decoupled setting? And the reasons to consider $\mathcal{F}$ and function query $h_t$ is the algorithm design and regret definition?
>
> In a decoupled setting, only one function $h_t$ is selected from the set of functions $\mathcal{F}$ to query at each step. This can improve sample and cost efficiency by reducing evaluations on functions that are less informative or more expensive to query. In contrast, in a coupled setting, all functions are evaluated at every step, which can be unnecessary and expensive. Bilevel problems often have multiple objective and constraint functions which highlights the benefits of decoupled setting. Decoupled settings have been explored in some constrained Bayesian optimization works as well [1, 2, 3].
>
> The function query selection is based on estimated regret comparison and a conditional reassignment for lower-level objective query. They were designed to lead to an instantaneous regret bound on the query point given by Lemma 3.8. In particular, the conditional reassignment is crucial to manage the uncertainty of the estimated lower-level optimal solution as we do not globally optimize the lower-level problem at any upper-level point unlike existing bilevel Bayesian optimization methods.
>
> [1] Gelbart, M. A., Snoek, J., & Adams, R. P. (2014). Bayesian optimization with unknown constraints. arXiv preprint arXiv:1403.5607.
>
> [2] Hern, J. M., Gelbart, M. A., Adams, R. P., Hoffman, M. W., & Ghahramani, Z. (2016). A general framework for constrained Bayesian optimization using information-based search. Journal of Machine Learning Research, 17(160), 1-53.
>
> [3] Nguyen, Q. P., Chew, W. T. R., Song, L., Low, B. K. H., & Jaillet, P. (2023). Optimistic Bayesian Optimization with Unknown Constraints. In The Twelfth International Conference on Learning Representations.
>
> >A more detailed comparison with existing bilevel optimization methods, particularly those offering theoretical guarantees or practical performance benchmarks, would provide a clearer picture of BILBO’s relative strengths and weaknesses.
>
> >For regret bound in Theorem 3.9, the authors discussed the relationship to constrained Bayesian optimization in Nguyen et al. (2023), is there any existing bilevel BO regret result that can be compared with?
>
> To the best of our knowledge, there are only two existing works on bilevel Bayesian optimization [4, 5], both of which do not provide theoretical guarantees, unlike BILBO. In addition, both existing works are not able to handle constraints while BILBO is able to. The performance benchmarks provided by both papers are based on the median converged fitness scores and median number of evaluations to convergence, with no indication on rate of convergence and confidence intervals, making it hard to compare directly.
>
> [4] Kieffer, E., Danoy, G., Bouvry, P., & Nagih, A. (2017). Bayesian optimization approach of general bi-level problems. In Proc. Genetic and Evolutionary Computation Conf. Companion.
>
> [5] Dogan, V., & Prestwich, S. (2023). Bilevel optimization by conditional Bayesian optimization. In Int. Conf. on Machine Learning, Optim., and Data Sci.
>
> >Including comparisons with state-of-the-art bilevel optimization algorithms, such as those from Kieffer et al. (2017) or Dogan & Prestwich (2023), would offer a more rigorous evaluation of BILBO’s performance.
>
> >In experiments, BILBO is compared with TrustedRandom and Nested. As described in Appendix C.1, the Nested baseline used a nested approach with SLSDP. But the related works mentioned, e.g.  Kieffer et al. (2017); Dogan & Prestwich (2023) are not included. Can you compare it with the associated works directly? If not, can you explain the reasons that BILBO cannot compared with the related work?
>
> We did not include direct comparisons with the two main related works [4, 5], as they did not provide sufficient implementation details or publicly available codes. We also found some potential errors in [5], where the calculation of mean conditional on optimal lower-level points seems to be incorrect. Both existing works use SLSQP for lower-level optimization, which we followed for our Nested baseline. Thus, the Nested algorithm in our experiments is a proxy to simulate the results of current methods to the best of our ability, and we believe this approach provides a reasonable benchmark given the constraints.

---

> > ### Author Response · Authors · 2024-11-17
> > **Official Comment by Authors (2/2)**
> >
> > >No code is provided alongside the paper. This omission hampers the reproducibility of the results and limits the ability of the research community to build upon or validate the findings presented.
> >
> > We will be providing code and data used upon acceptance of the paper. Seeds used for experiments in our paper will also be specified for reproducibility.
> >
> > >In Figure 1a, can you explain why BILBO has a sudden drop at around 150 queries? And what is the regret of BILBO after 150 queries (cannot observe the line afterwards)? The uncertainty level looks huge as well, I am wondering what would happen if you increase the number of runs (or change random seeds/initialisations, etc).
> >
> > We used an uniform grid discretization in our experiments, so the sudden drop is likely the difference between the regrets of the second most optimal point and optimal point, where the optimal point has no regret and thus exceeds the plot boundaries in logscale. The shaded areas are the 95\% confidence intervals. BILBO's confidence intervals may look huge because of this jump, but the confidence interval is mainly between the top two optimal points.
> >
> > >In experimental results, we observe Nested is generally similar to or worse than TrustedRand (which is only a random sampling), can you explain the possible reason? Does it suggest Nested is not a strong baseline (as it is from a previous work)?
> >
> > Indeed, Nested is sample inefficient as it requires global optimization of the lower-level problem at every upper-level query point. In comparison, TrustedRand randomly samples from trusted sets which provides estimates of lower-level optimal points without repeated lower-level optimizations. The observation that they are comparable and that TrustedRand occasionally outperforms Nested, shows that our trusted sets provide good approximations of lower-level optimal points and is effective in reducing the search space.
> >
> > We hope we have addressed your concerns with our discussion, especially on the motivation for decoupled setting and limitation of existing works, and improve your opinion of our work.

---

> > > ### Author Response · Authors · 2024-11-24
> > > **Seeking Your Valued Feedback on Our Response**
> > >
> > > We hope this message finds you well. We are following up on our recent response to your questions and reviews of our paper and would like to know whether our response adequately addressed all your concerns. Your feedback is crucial to us, so could you kindly take a moment to confirm if our rebuttal meets your expectations? We greatly appreciate the time and effort you have devoted to responding to us.

---

### Official Review · Reviewer_DxFV · 2024-11-04

**Soundness:** 3
**Presentation:** 4
**Contribution:** 2
**Rating:** 6
**Confidence:** 4

**Summary:**

This submission proposes a novel algorithm for bilevel Bayesian optimization that optimizes both upper and lower level problems jointly in a sample-efficient manner. The key idea of this algorithm is using confidence bounds to construct trusted sets of feasible and lower level optimal solutions. Theoretical studies show that trusted sets guarantee points with instantaneous regret bounds and sublinear regret bound is proven. Empirical studies are also provided.

**Strengths:**

1.	Bilevel Bayesian optimization is an important problem with many real-world applications in machine learning, economics, energy management, and investment.
2.	The proposed algorithm also works for the decoupled setting where both level functions may come from different simulators or experiments.
3.	Both theoretical and empirical studies are provided. Real-world experiments are reported to positively show the effectiveness of proposed algorithm.
4.	Overall the writing is good and easy to follow.

**Weaknesses:**

1.	I have concerns on novelty of this submission since all techniques in this paper are pretty standard. First, functions in both levels are modelled by Gaussian process and then trusted regions are defined using confidence bounds. Then theoretical results are derived from Srinivas et al., 2010. I cannot find challenges that are unique to the bilevel Bayesian optimization, or I might have missed something. Please clarify this point.
2.	In Theorem 3.9, theoretical results are given based on $|\mathcal{F}|,|\mathcal{X}|,|\mathcal{Z}|$, so all of them must be finite, which put more restrictions on problem setting.

**Questions:**

Optimization from two constructed sets can be very hard; see Step 7 of Algorithm 1. How do you implement this step while maintaining two sets in Step 17 efficiently? I guess only approximated solutions can be given.

---

> ### Author Response · Authors · 2024-11-17
> **Official Comment by Authors (1/2)**
>
> Thank you for appreciating our motivations for studying bilevel Bayesian optimization, the decoupled setting of our problem and our theoretical analysis and empirical experiments. We appreciate your valuable review and positive feedback on the clarity of our writing.
>
> We would like to address the following concerns:
>
> >I have concerns on novelty of this submission ... I cannot find challenges that are unique to the bilevel Bayesian optimization, or I might have missed something. Please clarify this point.
>
> The unique challenge in bilevel optimization is that the upper-level optimization is constrained by optimal lower-level solutions. The optimal lower-level solutions are unknown and requires optimization of a separate lower-level problem. A straightforward approach would be to globally optimize the lower-level problem at each upper-level query point, which is what existing Bayesian optimization-based approaches [1, 2] do. However, this is sample inefficient due to many unnecessary queries to the lower-level functions.
>
> We proposed a trust-region-like approach that maintains trusted sets of feasible and optimal lower-level solutions with high probability, and queries upper-level points without having to globally optimize the lower-level problem at any step. To our knowledge, our work is also the first to provide theoretical analysis and bounds on bilevel Bayesian optimization of blackbox functions.
>
> Our primary contributions include:
> - Trusted set of optimal lower-level solution $\mathcal{P}^+_t$ to approximate unknown optimal lower-level solutions, a challenge unique to bilevel optimization. We showed that points sampled from the trusted set $\mathcal{P}^+_t$ have upper-bounded lower-level objective regret, meaning they are good candidates for upper-level querying without have to globally optimize the lower-level problem repeatedly.
> - Function selection strategy involving comparison of estimated regrets and reassignment of lower-level query point given certain conditions. The reassignment is integral to manage uncertainties of the estimated lower-level solution, as we do not globally optimize the lower-level problem at any upper-level point. We showed that by following our function selection strategy at the query point, the instantaneous regret at the query point is upper bounded with high probability.
>
> [1] Kieffer, E., Danoy, G., Bouvry, P., & Nagih, A. (2017). Bayesian optimization approach of general bi-level problems. In Proc. Genetic and Evolutionary Computation Conf. Companion.
>
> [2] Dogan, V., & Prestwich, S. (2023). Bilevel optimization by conditional Bayesian optimization. In Int. Conf. on Machine Learning, Optim., and Data Sci.
>
> >First, functions in both levels are modelled by Gaussian process and then trusted regions are defined using confidence bounds. Then theoretical results are derived from Srinivas et al., 2010.
>
> To clarify, our theoretical results are not derived directly from [3], but only utilizes an adapted confidence bound Lemma as stated in Corollary 3.2, and we had represented the cumulative regret bound in terms of maximum information gain following [3] to facilitate comparison. These are common approaches in Bayesian optimization and present in several works [4, 5].
>
> The design of the trusted sets and  derivation of regret bounds for these sets build on these confidence bounds, but are entirely original work to tackle bilevel challenges. In addition, many existing Bayesian optimization works do not consider the decoupled setting where one function is selected for querying, and [3]'s work was on a single-objective, unconstrained optimization. Our proposed function query selection strategy which includes a conditional reassignment, leads to an upper regret bound on the query point and is a novel approach that addresses challenges associated with lower-level optimization of bilevel problems.
>
> [3] Srinivas, N., Krause, A., Kakade, S. M., & Seeger, M. (2009). Gaussian process optimization in the bandit setting: No regret and experimental design. arXiv preprint arXiv:0912.3995.
>
> [4] Xu, W., Jiang, Y., Svetozarevic, B., & Jones, C. (2023, July). Constrained efficient global optimization of expensive black-box functions. In International Conference on Machine Learning (pp. 38485-38498). PMLR.
>
> [5] Nguyen, Q. P., Chew, W. T. R., Song, L., Low, B. K. H., & Jaillet, P. (2023). Optimistic Bayesian Optimization with Unknown Constraints. In The Twelfth ICLR.
>
> >In Theorem 3.9, theoretical results are given based on $|\mathcal{F}|, |\mathcal{X}|, |\mathcal{Z}|$, so all of them must be finite, which put more restrictions on problem setting.
>
> These are likely fair assumptions to make, especially in real-world problems, where $\mathcal{F}$ will be a set of known experiments or simulators to run, and domains of $\mathcal{X}$ and $\mathcal{Z}$ can be obtained from domain knowledge. We have also added the assumption of finite domains of $\mathcal{X}$ and $\mathcal{Z}$ in the preliminaries to be clear.

---

> > ### Author Response · Authors · 2024-11-17
> > **Official Comment by Authors (2/2)**
> >
> > >Optimization from two constructed sets can be very hard; see Step 7 of Algorithm 1. How do you implement this step while maintaining two sets in Step 17 efficiently? I guess only approximated solutions can be given.
> >
> > We used uniform grid discretization for experiments in our paper, so we can represent the intersection of the two trusted sets as a set of points over which we find the maximum. We discussed several methods to improve the scalability of this approach to higher dimensions in our future work section, such as adaptive discretization which can be more computationally efficient.
> >
> > We hope our answers address your concerns and highlight the novel contributions of our work, which is the first bilevel Bayesian optimization with theoretical regret bounds and without inefficient repeated global optimization of the lower-level problem. We hope that this would improve your opinion of our paper.

---

### Official Review · Reviewer_nvZC · 2024-11-10

**Soundness:** 2
**Presentation:** 3
**Contribution:** 2
**Rating:** 3
**Confidence:** 3

**Summary:**

The paper investigates Bayesian Bilevel Optimization through a sampling-based approach. For general bilevel optimization problems, it provides a bound on the regret of the proposed algorithm and demonstrates its effectiveness across various synthetic and real-world problems.

**Strengths:**

- The writing is in general good and clear. I liked the motivation of Bilevel optimization.

- The paper performed extensive experiments on various datasets including some real-world examples. This is clearly a strength, though this could have been much more convincing if the paper is positioned to claim contributions on the empirical side, rather than the theoretical side.

**Weaknesses:**

The paper's introduction is somewhat misleading. Unlike typical optimization studies focused on the computational complexity of local-search algorithms, this paper addresses statistical complexity, assuming exhaustive search over a function hypothesis space. Consequently, it diverges from the usual challenges associated with noisy, constrained, and derivative-free settings in continuous stochastic optimization, making direct comparisons with gradient-based methods inappropriate.

Below are a few detailed comments:

- Line 59-60: What is "information flow"? This is not a well-defined term.

- Line 68-69: why functions are modelled using Gaussian Process? And this assumption appears abrupt.

- Gaussian process: notation is messy and discussion is hard to follow.

- At the beginning of Section 3, it is not clear what you exactly get from querying points.

- Line 155 "trusted sets": in my first read, this was very confusing. I think the paper should have been much clearer about the interaction protocol, the fact that exhaustive search over hypothesis class is okay, and it is not going to be about convergence of local-search methods.

- Line 301-302: What is "information gain"? Define it precisely.

While the extensive experiments in Section 4 are impressive, it remains unclear how to evaluate this section if the main contribution is claimed on methodological rather than real-world applicability.

**Questions:**

See weaknesses.

---

> ### Author Response · Authors · 2024-11-17
>
> Thank you for finding our motivation of bilevel optimization clear and appreciating our extensive experiments. We appreciate your valuable review.
>
> We would like to address the following concerns:
>
> >Unlike typical optimization studies focused on the computational complexity of local-search algorithms, this paper addresses statistical complexity, assuming exhaustive search over a function hypothesis space. Consequently, it diverges from the usual challenges associated with noisy, constrained, and derivative-free settings in continuous stochastic optimization, making direct comparisons with gradient-based methods inappropriate.
>
> >Line 68-69: why functions are modelled using Gaussian Process? And this assumption appears abrupt.
>
> >Gaussian process: notation is messy and discussion is hard to follow.
>
> Our paper proposes bilevel optimization via Bayesian optimization, which is a global optimization algorithm commonly used in noisy and derivative-free settings. Gaussian processes are also commonly used to model functions in Bayesian optimization due to their probabilistic properties, and the Gaussian process notation and definition are standard and commonly found in Bayesian optimization literature [1, 2, 3]. Nevertheless, we shifted the closed-form posteriors to the appendix to enhance readability.
>
> [1] Williams, C. K., & Rasmussen, C. E. (2006). Gaussian processes for machine learning (Vol. 2, No. 3, p. 4). Cambridge, MA: MIT press.
>
> [2] Brochu, E., Cora, V. M., & De Freitas, N. (2010). A tutorial on Bayesian optimization of expensive cost functions, with application to active user modeling and hierarchical reinforcement learning. arXiv preprint arXiv:1012.2599.
>
> [3] Srinivas, N., Krause, A., Kakade, S. M., & Seeger, M. (2009). Gaussian process optimization in the bandit setting: No regret and experimental design. arXiv preprint arXiv:0912.3995.
>
> >At the beginning of Section 3, it is not clear what you exactly get from querying points.
>
> Querying means to evaluate or observe a function at a specific point. This is also common terminology in Bayesian optimization [2].
>
> >Line 155 "trusted sets": in my first read, this was very confusing. I think the paper should have been much clearer about the interaction protocol, the fact that exhaustive search over hypothesis class is okay, and it is not going to be about convergence of local-search methods.
>
> We do not do an exhaustive search over hypothesis class nor local-search methods. Our work is on Bayesian optimization with Gaussian processes as surrogate models, where we select a query point and function query at each step to efficiently find the optimal solution. We construct trusted sets using confidence bounds from Gaussian processes to iteratively reduce the search space.
>
> >Line 59-60: What is "information flow"? This is not a well-defined term.
>
> In this context, we are using the term "information" informally. When optimization for each level is carried out separately, there is no information exchange (e.g. uncertainties) between upper- and lower-level problems.
>
> >Line 301-302: What is "information gain"? Define it precisely.
>
> We have included the definition of maximum information gain in the preliminaries and its upper bound for commonly used kernels in the appendix of the updated paper.
>
> We hope this clarifies and improves your opinion of our paper.

---

> > ### Author Response · Authors · 2024-11-24
> > **Seeking Your Valued Feedback on Our Response**
> >
> > We hope this message finds you well. We are following up on our recent response to your questions and reviews of our paper and would like to know whether our response adequately addressed all your concerns. Your feedback is crucial to us, so could you kindly take a moment to confirm if our rebuttal meets your expectations? We greatly appreciate the time and effort you have devoted to responding to us.

---

### Author Response · Authors · 2024-12-04

We thank the reviewers for taking the time to provide thoughtful feedback. We are encouraged that they found our work to be novel (xVUG, 7vtx), our bilevel optimization problem to be important (xVUG, DxFV), and our writing clear and well-structured (nvZC, DxFV, 7vtx, pSWd). We are pleased that they acknowledged our approach of concurrent optimization of both levels enhances sample efficiency (xVUG, 7vtx), and recognized our work as the first theoretically guaranteed approach for bilevel Bayesian optimization (pSWd). We are also grateful that they identified our theoretical analysis and practical experiments as strengths of our work (DxFV, 7vtx), which includes a sublinear regret bound (xVUG), infeasibility declaration (xVUG), and comprehensive proofs (pSWD).

We appreciate the opportunity for this discussion, where we have elaborated on significant challenges of bilevel Bayesian optimization of blackbox functions and our novel methods to overcome them, successfully addressing novelty concerns (DxFV, pSWd). We have incorporated these discussions and feedback into the updated paper, with no outstanding or major issues.

---

### Meta-Review · Area_Chair_qff6 · 2024-12-22

**Metareview:**

This submission proposes a new algorithm for bilevel Bayesian optimization. The key idea is to use confidence bounds to construct trusted sets of feasible and lower-level optimal solutions. The authors establish a sublinear regret bound and validate the new algorithm through experiments

This is a borderline submission. Out of the five designated reviewers, four reviewers (who are familiar with Bayesian optimization) gave a score of 6. The other reviewer, who is familiar with optimization in general but not that familiar with Bayesian optimization, gave a score of 3.

Although the four reviewers who are familiar with Bayesian optimization all lean towards accepting this paper, none of them seem to be enthusiastic about its acceptance. In particular, some of them mentioned explicitly in their reviews that the technical novelty of this submission could be limited.

The main criticism from the reviewer who gave a score of 3 is the clarity of the introduction part of the paper. Although this criticism is from someone who is not that familiar with Bayesian optimization, such weakness should be taken into consideration since ICLR is a large conference and the audiences have diverse backgrounds. Moreover, given that this submission might suffer from the issue of technical novelty, it must have done a great job in terms of clarity to ensure acceptance.

Given all the factors mentioned above and the high standards of ICLR, I lean towards rejecting this submission.

**Additional Comments On Reviewer Discussion:**

The reviewers raised concerns regarding the technical novelty of the paper, clarity of the introduction part of the paper, as well as the motivations for certain design choices. Although the authors provided responses which addressed some of those concerns, concerns regarding clarity of the introduction part of the paper remain.

---

### Decision · Program_Chairs · 2025-01-22

Reject